# Beware of Overestimated Decoding Performance Arising from Temporal Autocorrelations in Electroencephalogram Signals

## Abstract

Researchers have reported high decoding accuracy (>95%) using non-invasive Electroencephalogram (EEG) signals for brain-computer interface (BCI) decoding tasks like image decoding, emotion recognition, auditory spatial attention detection, etc. Since these EEG data were usually collected with well-designed paradigms in labs, the reliability and robustness of the corresponding decoding methods were doubted by some researchers, and they argued that such decoding accuracy was overestimated due to the inherent temporal autocorrelation of EEG signals. However, the coupling between the stimulus-driven neural responses and the EEG temporal autocorrelations makes it difficult to confirm whether this overestimation exists in truth. Furthermore, the underlying pitfalls behind overestimated decoding accuracy have not been fully explained due to a lack of appropriate formulation. In this work, we formulate the pitfall in various EEG decoding tasks in a unified framework. EEG data were recorded from watermelons to remove stimulus-driven neural responses. Labels were assigned to continuous EEG according to the experimental design for EEG recording of several typical datasets, and then the decoding methods were conducted. The results showed the label can be successfully decoded as long as continuous EEG data with the same label were split into training and test sets. Further analysis indicated that high accuracy of various BCI decoding tasks could be achieved by associating labels with EEG intrinsic temporal autocorrelation features. These results underscore the importance of choosing the right experimental designs and data splits in BCI decoding tasks to prevent inflated accuracies due to EEG temporal correlations. The watermelon EEG dataset collected in this work can be obtained at Zenodo: https://zenodo.org/records/11238929, and all the codes of this work can be obtained in the supplementary materials.

## 1 Introduction and related works

A brain-computer interface (BCI) is a type of human-machine interaction that bridges a pathway from the brain to external devices [1]. Electroencephalogram (EEG) has emerged as a valuable tool for BCI because of its high time resolution, low cost, and good portability [2], and algorithms of neural decoding from EEG signals play a role in its practical applications. Recently, deep learning methods have been developed widely for various EEG decoding tasks, and high decoding accuracy was reported. For example, in the task of decoding image classes with EEG recordings, when subjects were required to watch images of different classes, a decoding accuracy of 82.90% was reported for the 40-way classification by Spampinato et al. [3]. With their EEG dataset, subsequent

studies reported a higher decoding accuracy (98.30%, [4]), high performance on image retrieval, and even image generation from EEG [5, 6, 7].

However, it remains unclear what kind of EEG features are learned by the DNN-based models. Some researchers have posited that the high decoding accuracy on the image-evoked EEG dataset was attributed to the block-design paradigm during EEG recording [8, 9, 10], in which 50 images with the same class label were presented to the subject continuously in one block, and the 40 image-classes were presented as 40 separate blocks. Due to the existence of temporal autocorrelation of EEG signals, i.e., the temporally nearby data is more similar than the temporally distal [11, 12, 13, 14], the models could learn the block-related features rather than the image-related.

To verify their concerns, Li et al. [8] recorded EEG with two experimental designs: block design and rapid-event design. For the rapid-event design, images across the 40 classes were presented alternately and randomly. When the same DNN model was used, it was found that the decoding accuracy was close to Spampinato et al. [3] with the block-design EEG data, but it was dramatically decreased to the chance-level (2.50%) with the rapid-event design data. Subsequent work also confirmed the low decoding accuracy for EEG recorded with rapid-event design [9, 10]. However, Palazzo et al. [15] proposed that temporal autocorrelations only play a marginal role in EEG decoding tasks because they found that EEG data recorded during rest periods (temporal proximity to adjacent blocks) could not be successfully classified as the preceding block label or the succeeding block label. They also argued that the rapid-event design seemed to weaken the image-related neural responses due to the possible cognitive load and fatigue effect compared to the block design. Some researchers [15, 16, 17, 18] pointed out that block design is essential because humans tend to react more consistently and respond faster when conditions are presented in blocks [19, 20]. Wilson et al. [18] advised that classification work that decodes from block design datasets is the most suitable approach until advances are made to reduce noise.

Although the pitfall of overestimated decoding accuracy has been mainly discussed in image neural decoding tasks, we noticed that similar pitfalls might also exist in various EEG decoding tasks such as in auditory spatial attention detection (ASAD) tasks [21, 22, 23, 24], which involves decoding the subjects auditory attention locus from neural data, and in emotion recognition task [25, 26, 27], which involves recognizing the subjects emotion type from neural data. Researchers have also found that splitting a continuous EEG from a specific experimental condition into training and test sets would bring higher decoding accuracy in epilepsy detection tasks [28], motor imagery decoding tasks [29], and so on. All those high decoding accuracy works share the common characteristic: continuously recorded EEG data of a specific class (condition) label are divided into training and test sets (see the top-left of Figure 1).

Although some studies have mentioned the overestimated decoding accuracy and tried to remind the possible pitfall [8, 30], it is difficult to discriminate the influence of the inherent temporal autocorrelation in EEG signals due to the coupling of stimuli-driven neural responses and the temporal autocorrelations. More importantly, due to the lack of an effective formalization, there is not an adequate explanation of how models utilize temporal autocorrelation features for decoding. Furthermore, their concerns only focused on one specific decoding task, and the results and conclusions cannot be generalized to general BCI decoding tasks.

In this work, the pitfall of various EEG decoding tasks was formulated with a unified framework. To completely decouple the temporal autocorrelation features from stimuli-driven neural responses, EEG data were collected from 10 watermelons in this work to construct "Watermelon EEG". This method is known as phantom EEG in previous studies [31, 32, 33, 34, 35, 36], and the EEG data exclude stimulus-driven neural responses while reserving the temporal autocorrelation features. For comparison, a human EEG dataset was also adopted. The watermelon EEG and human EEG were reorganized into three classic neural decoding EEG datasets following their EEG experimental paradigm: image classification (CVPR, [3]), emotion classification (DEAP, [37]), and auditory spatial attention decoding (KUL, [38]), resulting in six EEG datasets. A sample CNN-based decoding model was used to complete the decoding tasks with the corresponding EEG dataset, and the experimental results revealed that:

1. When the pitfall was formulated with a unique framework, and the temporal autocorrelation was defined as domain features, high decoding accuracy of various BCI decoding tasks could be achieved by associating labels with EEG intrinsic temporal autocorrelation features.

2. The pitfall exists not only in classification but also widely in EEG-image joint training without explicit labels and even image generation.

3. Splitting a continuous EEG with the same class label into training and test sets should never be used in future BCI decoding works.

## 2 Method

The section is organized by: the pitfall is formulated in Subsection 2.1, and the datasets used are introduced in Subsection 2.2. Then, the methods to finish different classification tasks are introduced in Subsection 2.3, and joint training and image generation from EEG are introduced in Subsection 2.4. Some implementation details and statistical analysis method are described in Subsection 2.5.

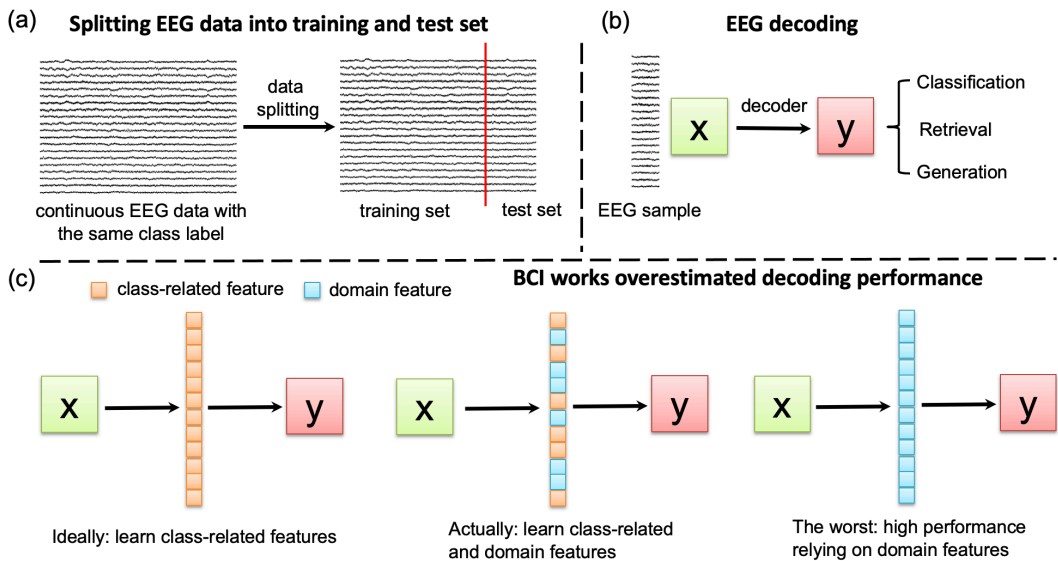

Figure 1: Overestimated decoding performance in BCI works. (a) Continuous EEG data in a certain experimental condition (with the same class label) are split into training and test sets for decoder training and evaluation. (b) With the test EEG sample input, the decoder gives output in the forms of classification, retrieval, and generation. (c) Decoders may use both domain features or class-related features for decoding.

### 2.1 Problem Formulation

In some BCI works on domain generalization [39], all EEG data from a dataset [40] or from a subject [41] are usually regarded as a domain to emphasize EEG pattern distribution differences between datasets or subjects. Adopted from this concept, we regard a period of continuous EEG data with the same class label as a domain. In some BCI works [3, 4, 21, 22, 23, 24, 25, 26, 27], researches segment the EEG data from the same domain into samples and further split the samples into training and test data (as shown in Figure 1a) and complete decoding task, such as classification, retrieval and generation (as shown in Figure 1b). In these cases, the models used in these works would learn the coupled features containing the class-related feature and domain feature (as shown in the middle of the Figure 1c). The underlying assumption of these works is that the domain feature plays only a margin role in EEG decoding tasks as shown in the left of the Figure 1c. However, we assumed that the domain feature contributes to the high decoding accuracy as shown in the right of the Figure 1c, which is the pitfall we mentioned in Section 1.

To validate our assumption, we need to formulate the pitfall. Denote $D$ as the domain set, and each domain $d \in D$ contains many samples. We use $S^d$ to denote the sample set of the domain $d$. The notation $x_i^d$ represents the $i$-th sample (e.g., a 0.5-second EEG data corresponding to watching a specific image) of domain $d$, which is associated with class $y_i^d$ (e.g., the class label panda of the

watched image). Considering the temporal autocorrelation of the EEG data, the domain features of data within the same domain are more similar, while the domain features of data in different domains are more distinct.

For EEG decoding tasks, we assume the data is generated from a two-stage process. First, each domain is modeled as a latent factor $z$ sampled from some meta domain distribution $p(\cdot)$. Second, each data sample $x$ is sampled from a sample distribution conditioned on the domain $z$ and class $y$:

$$z \sim p(\cdot), x \sim p(\cdot|z, y) \tag{1}$$

Given the sample $x$, the aim of a specific EEG decoding task is to uncover its true class label using the posterior $p(y|x)$. The quantity can be factorized by the domain factor $z$ as,

$$p(y|x) = \int p(y, z|x)dz = \int p(y|x, z)p(z|x) \tag{2}$$

When we use the Watermelon EEG dataset or use a dataset that is completely unrelated to the current task (e.g., decoding images from an auditory EEG dataset), the class-related feature has none possibility to exist in EEG samples. In this condition, $p(y|x, z) = p(y|z)$ and the equation (2) can be modified as:

$$p(y|x) = \int p(y, z|x)dz = \int p(y|z)p(z|x) \tag{3}$$

The assumption of this work is that the model could also deduce $p(y|x)$ by learning $p(y|z)$ and $p(z|x)$ even there is none class-related feature exists. In other words, we assumed that it could also achieve high decoding accuracy on different EEG decoding tasks when using the Watermelons EEG dataset.

## 2.2 Dataset

**Watermelon EEG Dataset** Ten watermelons were selected as subjects. EEG data were recorded with a NeuroScan SynAmps2 system (Compumedics Limited, Victoria, Australia), using a 64-channel Ag/AgCl electrodes cap with a 10/20 layout. An additional electrode was placed on the lower part of the watermelon as the physiological reference, and the forehead served as the ground site (see Appendix A.1 for photography). The inter-electrode impedances were maintained under 20 kOhm. Data were recorded at a sampling rate of 1000 Hz. EEG recordings for each watermelon lasted for more than 1 hour to ensure sufficient data for the decoding task. We refer to the dataset consisting of EEG recordings of 10 watermelons as the Watermelon EEG Dataset.

**SparrKULee Dataset** SparrKULee dataset[42] is a speech-evoked EEG dataset from the KU Leuven University containing 64-channel EEG recordings from 85 participants, each of whom listened to 90-150 minutes of natural speech. We used this dataset because EEG recordings were longer than 1 hour to ensure a sufficient amount of data for each subject. To match the number of subjects in the Watermelon EEG Dataset, EEG data from 10 subjects (ID: Sub7-Sub16) from the SparrKULee Dataset were used.

**Dataset reorganization and dataset segmentation** The term "reorganization" refers to segmenting continuous EEG into samples and assigning each sample a class label and a domain label according to the referenced experimental design. Here, we follow the experimental designs of three classical published EEG datasets to reorganize the Watermelon EEG Dataset and SparrKULee Dataset. These three datasets were collected respectively for image decoding, emotion recognition, and ASAD tasks.

For the image decoding task, we referred to the experimental design of the CVPR dataset [3]. For the CVPR dataset, 40 classes of images were presented in a block-design paradigm. Specifically, 50 different images of the same class were presented continuously in a block, with each image lasting for 0.5 second, resulting in 40 blocks of presentation for each subject. The 0.5-second length EEG data of the same class were split into training, validation, and test sets in a ratio of 8:1:1 [4, 3]. Following this experimental design and dataset segmentation, we segment continuous EEG from

the Watermelon EEG Dataset and SparrKULee Dataset into blocks and assign a unique class label and a unique domain label for each block. The interval between adjacent blocks is set to 10 seconds to match the rest time of the subjects during the EEG recording in the CVPR dataset. Then, EEG data in each block are further segmented into 50 0.5-s length samples. Since the EEG data in the CVPR dataset has 128 channels, we replicated our 64-channel EEG in the channel dimension. The reorganized datasets for Watermelon Dataset and SparrKULee Dataset are called WM-CVPR and SK-CVPR, respectively. Here, we use the "A-B" naming format, where the left side of "-" represents the source dataset (WM: watermelon dataset, SK: SparrKULee Dataset), and the right side of "-" represents the dataset of which the experimental design is referenced. For the emotion recognition task and ASAD task, the DEAP dataset and the KUL dataset are used as the referenced dataset, resulting in WM-DEAP, SK-DEAP, WM-KUL, and SK-KUL. More details for reorganization can be found in Appendix A.2.

## 2.3 Classification tasks

**Model.** To demonstrate that domain features are strong and easy to be learned by the network, we used a simple CNN (or some parts of this CNN) to complete all classification tasks mentioned in this work. The CNN network includes a layer-norm layer, a 2D-convolutional layer (output channel: 100), an averaging pooling layer, and two fully connected layers. The kernel size of the 2D-convolutional layer depends on the channel number and sampling frequency of the input EEG. The node number of the output fully connected layer depends on the number of classes.

**Decoding the domain feature** To demonstrate that the model can predict the domain factor $z$ from EEG input sample $x$, which relates to learning posterior $p(z|x)$, a domain label classification was adopted on the six datasets (i.e., WM-CVPR, WM-DEAP, WM-KUL, SK-CVPR, SK-DEAP and SK-KUL dataset) with a simple CNN classifier. The splitting strategy leave-samples-out was used, which means that all sample were randomly split into training set, validation set and test set. The outputs after the averaging pooling layer were selected as domain feature representation, and t-SNE was utilized for dimensionality reduction and visualization.

**Decoding the class label from the domain feature** To demonstrate that the model can predict the class label $y$ from the domain factor $z$, which relates to learning posterior $p(y|z)$, a class label classification was adopted on the four datasets (classification on the WM-CVPR dataset and SK-CVPR dataset are unnecessary since domain labels and class labels are one-to-one correspondence) using a single network with two linear layers and an intermediate sigmoid function.

**End-to-end classification** To demonstrate that the model can predict the class label $y$ from the EEG input sample $x$ directly when samples in the training set and test set are from common domains, a class label classification was adopted on the six datasets with the simple CNN classifier. The splitting strategy leave samples out was used. Classification on the WM-CVPR dataset and SK-CVPR dataset is the same since domain labels and class labels in the two datasets are one-to-one correspondence. To demonstrate that the model indeed used the domain feature to complete the end-to-end classification, the splitting strategy leave domains out was used on the four datasets (i.e., WM-DEAP, WM-KUL, SK-DEAP, and SK-KUL dataset) in which samples in the same domain only appear in the training set or the test set.

**Zero-shot classification** In a recent work [4], EEG data from 34 classes within the CVPR2017 dataset were used to train an EEG encoder, and the remaining 6 unseen classes were used for testing. The results showed that features of different unseen classes clustered in distinct groups on the two-dimensional t-SNE plane. Similar analyses were conducted on the SK-CVPR and WM-CVPR datasets. Six classes were selected for testing, and the remaining 34 classes were for training. The simple CNN was used to predict class labels from input EEG samples, and the outputs from the average pooling layer were chosen as the EEG feature representation. Two strategies were employed for selecting the 6 test classes: random selection and first-six selection. For random selection, the 6 test classes are randomly chosen from the 40 classes. For the first-six selections, the first presented 6 classes in the EEG experiment are chosen. During the test stage, since the training set does not include classes corresponding to the test EEG data, the model could not give the corresponding labels and could only output the most probable classes among the 34 seen during training. Therefore, we proposed two evaluation metrics:$Acc_{near}$ and $Acc_{7th}$. $Acc_{near}$ represents the proportion of EEG data classified into temporally adjacent classes, while $Acc_{7th}$ represents the proportion classified into the category presented seventh in time.

## 2.4 Joint training and image generation

To demonstrate that the model can utilize domain features to accomplish retrieval and generation besides classification, EEG-image joint training and image generation on WM-CVPR and SK-CVPR were conducted.

**Joint training** In the EEG-image joint training, a pre-trained image encoder was typically utilized to extract image representation, while an EEG encoder was employed to extract EEG features to align with the image representation. During the decoding process, a retrieval task was applied. Specifically, given a test EEG sample and a collection of images containing the target and the non-target. The image representation was reconstructed from the EEG with the EEG encoder. The similarity between the reconstructed image representation and all candidate image representations in the collection is calculated. The decoded output image is selected based on the ranking of these similarities. Usually, the Top-k accuracy and normalized Rank accuracy are used as evaluation metrics. In this work, the simple CNN described in Subsection 2.3 is used as an EEG encoder. The detailed implementation can be found in Appendix A.3.

**Image generation** The image generation aims to generate images seen by the subjects from their EEG data. This task commonly uses a two-stage process: EEG encoding and image generation. In the EEG encoding stage, a model is built to encode EEG data into a latent representation. In the image generation stage, a pre-trained image generator is used. The generator is fine-tuned with EEG representation and corresponding images. In this work, the EEG data are first encoded into image representation with a simple CNN described in Subsection 2.3. Following previous work[43], a latent diffusion model conditioned on image representation was used. The metric of n-way top-k accuracy was used for evaluating the semantic correctness of generated images [44]. The detailed implementation can be found in Appendix A.4.

## 2.5 Implement details

The neural networks were implemented with the Pytorch and trained on a single high-performance computing node with 8 A800 GPU. For the classification task, the AdamW [45] optimizer was employed to minimize the cross-entropy loss function with a learning rate of $10^{-3}$. For the joint training and image generation, the AdamW optimizer was used with a learning rate of $10^{-3}$ and $5 \times 10^{-4}$ for each task respectively. More details can be found in our codes. All the experiments mentioned in this work were trained within the subjects (i.e., models were trained for each subject respectively) except special annotation (unseen subject decoding results were only presented in Appendix A.5). For statistical analysis, the one-sample t-test was used to check whether the reported results were significantly higher than the chance level. Bonferroni correction was used to adjust the $p$-value. A $p$-value of 0.05 or lower was considered statistically significant.

# 3 Results

## 3.1 Classification tasks

The results shown in Table 1 present that classification accuracy in domain label classification and class label classification are all significantly above the chance level. This shows that the domain feature can be extracted effectively with a simple CNN, and the label class can be decoded from the extracted domain features or from EEG directly. In contrast, the decoding accuracy drops to the chance level when using the splitting strategy leave-domains-out, further supporting domain feature-induced high decoding accuracy. The standard error of the mean calculated over the subjects level is reported for accuracy in this work.

Figures 2a and 2b show the t-SNE plot for domain label classification and end-to-end class label classification. As shown in Figure 2a, 8 distinct clusters exist, each corresponding to one domain. In Figure 2b, 8 distinct clusters also exist, with four corresponding to class label 1 and the other four corresponding to class label 2. This indicates that the high decoding accuracy results from associating class labels with domain features.

Table 1: Classification accuracy (%) on the six datasets. DLC is for domain label classification. TLC-DF is for class label classification from domain features. TLC-EEG is for end-to-end class label classification. TLC-EEG-woDO is for class label classification direct from EEG when samples in the training set and test set are from different domains.

| | WM-CVPR | WM-DEAP | WM-KUL | SK-CVPR | SK-DEAP | SK-KUL |
|---|---|---|---|---|---|---|
| DLC | $88.78 \pm 4.95$ | $96.98 \pm 0.76$ | $99.99 \pm 0.01$ | $69.83 \pm 2.98$ | $72.70 \pm 1.36$ | $100.00 \pm 0.00$ |
| DLC (chance level) | 2.50 | 2.50 | 12.50 | 2.50 | 2.50 | 12.50 |
| TLC-DF | - | $92.77 \pm 1.31$ | $100.00 \pm 0.00$ | - | $76.19 \pm 1.80$ | $100.00 \pm 0.00$ |
| TLC-EEG | $88.78 \pm 4.95$ | $88.74 \pm 3.26$ | $82.74 \pm 6.44$ | $69.83 \pm 2.98$ | $74.44 \pm 2.76$ | $93.34 \pm 2.01$ |
| TLC-EEG-woDO | - | $24.67 \pm 2.31$ | $49.97 \pm 4.67$ | - | $25.34 \pm 1.85$ | $59.32 \pm 4.07$ |
| TCL (chance level) | 2.50 | 25.00 | 50.00 | 2.50 | 25.00 | 50.00 |

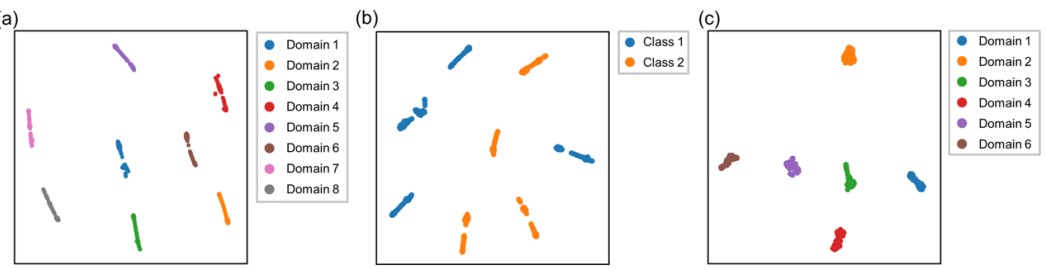

Figure 2: t-SNE plot for (a) domain label classification, (b) end-to-end class label classification, and (c) zero-shot class label classification

The experimental results for zero-shot classification are displayed in Table 2. It can be observed that the model tended to classify test samples into temporally adjacent classes. Figure 2c shows the t-SNE visualization of the unseen EEG features extracted from the decoder. Despite being unseen, different domains of features clustered in distinct groups. This suggests that the decoder just learned to extract EEG domain features during training and distinguish unseen EEG responses from the domain features.

Table 2: Zero-shot EEG classification accuracy (%) on WM-CVPR and SK-CVPR datasets.

| | WM-CVPR first-six | WM-CVPR random | SK-CVPR first-six | SK-CVPR random |
|---|---|---|---|---|
| $Acc_{near}$ | - | $79.43 \pm 5.61$ | - | $78.00 \pm 5.66$ |
| $Acc_{7th}$ | $69.60 \pm 10.64$ | $6.73 \pm 3.24$ | $77.03 \pm 11.32$ | $0.87 \pm 0.82$ |

### 3.2 Joint training and image generation

For EEG-image joint training, Table 3 displays the accuracy for the retravel task on the test set. The table shows that, for both types of loss functions, decoding accuracy is far above the chance level, demonstrating that the model can utilize domain features to align EEG with image features. Table 3 Result for joint training on WM-CVPR and SK-CVPR with a loss function of cosine similarity (CS) or InfoNCE.

Table 3: Accuracy (%) for joint training on WM-CVPR and SK-CVPR with a loss function of cosine similarity (CS) or InfoNCE.

| | WM-CVPR | | SK-CVPR | | Chance level |
|---|---|---|---|---|---|
| | CS loss | InfoNCE loss | CS loss | InfoNCE loss | |
| Top1 Acc | $81.40 \pm 9.25$ | $90.15 \pm 5.45$ | $80.70 \pm 0.60$ | $79.70 \pm 0.92$ | 2.50 |
| Top5 Acc | $90.65 \pm 5.82$ | $98.56 \pm 1.09$ | $88.86 \pm 1.03$ | $92.39 \pm 0.38$ | 12.50 |
| Rank Acc | $95.87 \pm 2.51$ | $99.42 \pm 0.38$ | $95.20 \pm 0.24$ | $98.09 \pm 0.07$ | 50.00 |

For image generation, Table 4 displays the n-way top-k accuracy for the generated images on the WM-CVPR and SK-CVPR datasets. The metrics are significantly above the chance level, indicating

that the generated images have correct semantics. Figure 3 shows some generated images on the WM-CVPR dataset. As shown in the figure, the model can exactly generate the correct images. The results on EEG-image joint training and image generation show that in addition to classification tasks, retrieval, and generation can also achieve high performance by leveraging domain features shared by the test and training sets.

Table 4: Accuracy (%) for semantic correctness. The repeated times N was set to 50.

| - | Top-1/50-way | Top-5/50-way | Top-1/100-way | Top-5/100-way |
|---|---|---|---|---|
| WM-CVPR | $26.77 \pm 3.37$ | $46.44 \pm 4.60$ | $21.64 \pm 2.89$ | $38.11 \pm 4.30$ |
| SK-CVPR | $25.04 \pm 0.93$ | $43.61 \pm 0.88$ | $20.37 \pm 0.91$ | $35.35 \pm 0.89$ |
| Chance | 2.00 | 10.00 | 1.00 | 5.00 |

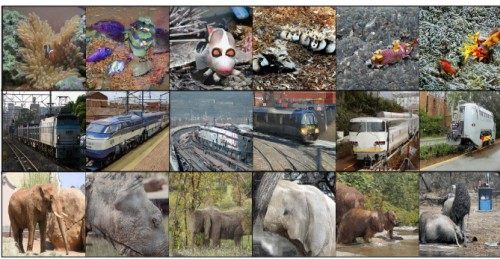 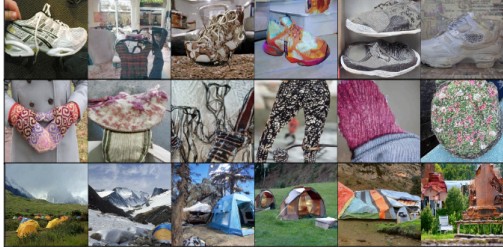

Figure 3: EEG-generated image from a typical watermelon subject, where the first column of each panel represents the real images "watched" by the watermelon subject, and the following five columns show the images generated by the model.

## 4 Discussion

### 4.1 Relying on the domain features for EEG decoding

While many works on EEG decoding have reported high-performance results, we proposed that some of these high-performance may rely on temporal autocorrelation of EEG data. The pitfall may involve different EEG decoding tasks. To clarify this pitfall, the concept of domain was adopted to describe the temporal autocorrelation of a continuous EEG with the same label. EEG data were collected from watermelon as the phantom to exclude the contribution of stimuli-driven neural responses to decoding results. The results showed that a simple CNN network could well learn domain features from EEG data and could associate class labels with domain features.

To avoid the pitfalls, a feasible approach is to adopt a reasonable data-splitting strategy to avoid training and test sets sharing the common domain features, i.e., a leave-domains-out splitting strategy. For instance, a leave-subjects-out data-splitting strategy can be adopted, which entails designating the data from certain participants for training and data from others for testing. Alternatively, for datasets that do not follow a block design, a leave-trials-out strategy may be applied. Prior research has consistently demonstrated that employing a leave-subjects-out splitting strategy precipitates a notable decline in decoding performance [46]. In some cases, it has been reported that decoding accuracy dropped to the chance level [47, 8]. The prevalent interpretation is that inter-individual variability [46] hampers the generalizability across different subjects. However, we posit that the observed decrement in decoding accuracy is attributable to model overfitting to domain features. Although the leave-subjects-out partitioning strategy is designed to prevent the leakage of domain features, the presence of these domain features in the training set can still lead the model to inadvertently exploit them to differentiate between categories during the training phase. The methods and results further support the conclusion can be found in Appendix A.5

Palazzo et al. [15] proposed that the EEG temporal correlation related to baseline drift could be alleviated by high-pass filtering. However, our further experiment proved that the domain feature still exists and that high decoding accuracy could be achieved in any frequency band (see Appendix A.6). We argue that the focus should not be exclusively on the elimination of EEG autocorrelation through

filtering. Instead, greater emphasis should be placed on the experimental paradigms of EEG recording and the methods employed for dataset splitting. By addressing these aspects, we can proactively prevent the overestimated decoding accuracy arising from EEG temporal autocorrelations.

It is worth noting that we do not want to create an illusion that all BCI works utilize EEG temporal autocorrelation features for decoding. In fact, there are many works that do not rely on EEG temporal autocorrelation features for decoding in image decoding [48, 49, 50] emotion recognition [51], sleep detection [40, 41] and ASAD [52]. These works demonstrated the feasibility of various BCI tasks.

## 4.2   Potential sources of domain features

In this work, we have demonstrated the existence of EEG temporal autocorrelation in the watermelon EEG, which consists of no neural activities, and in the human EEG data. Li et al. [8] believed the model decodes by utilizing the baseline drift in the CVPR2017 dataset. They found that when the EEG data is filtered with a bandpass filter, the decoding accuracy dropped greatly. Palazzo et al. [15] also claimed that temporal correlation was strong only in low frequency. However, we have demonstrated in Appendix A.4 that the domain feature still exists and that high decoding accuracy can be achieved in any frequency band. In addition to baseline drift, some neuroscience works have shown that temporal autocorrelation existed in neural oscillation, which could be reflected in EEG in various frequency bands. This is referred to as Long-Range Temporal Correlations (LRTC) in neuroscience research [11, 12, 13, 14]. Linkenkaer-Hansen et al. [13] first calculated the LRTC in resting-state EEG data. They found that spontaneous alpha, mu, and beta oscillations result in significant LRTC for at least several hundred seconds during resting conditions. Subsequent neuroscience research further demonstrated that significant LRTC exists in the theta [11] and gamma [12] bands. While baseline drift can be removed through filtering, the frequency range of the LRTC overlaps with the frequency range of stimuli-driven neural responses, making it impossible to remove this domain feature through filtering. Temporal correlation analysis on human EEG in the SparrKULee Dataset showed the existence of strong LRTC in all frequency bands, and the LRTC in a narrowband is sufficient to complete the corresponding decoding task. The methods and results further support the conclusion can be found in Appendix A.7.

## 4.3   Limitation and future work

Although direct evidence of overestimated decoding accuracy attributable to domain feature across various brain-computer interface (BCI) tasks have been provided in the current work, no solution has been proposed to mitigate overfitting to domain features in the training set. Some works have already used domain adaptation [2, 53, 54] or domain generalization [40, 41] method to improve decoding accuracy under leave-subjects-out data splitting in BCI tasks. This may also help alleviate the adverse effects of domain features on decoding tasks. It is also noteworthy to highlight the remarkable efficacy of large-scale EEG model in various BCI decoding tasks [55, 56, 57]. Given that domain features are pervasive in extensive EEG datasets and do not necessitate manually annotated labels, self-supervised pre-trained large EEG models may be especially adept at discerning and neutralizing domain features, thereby facilitating more robust and generalizable decoding performance.

# 5   Conclusion

In this work, the "overestimated decoding accuracy pitfall" in various EEG decoding tasks is formulated in a unified framework by adopting the concept of "domain". Some typical EEG decoding tasks (image decoding, emotion recognition, and auditory spatial attention detection) are conducted on the self-collected watermelon EEG dataset. The results showed that EEG data from different domains have distinctive domain features induced by EEG temporal autocorrelations. Using the inappropriate data partitioning strategy, high decoding accuracy is achieved by associating class labels with domain features. The results will draw attention to the high decoding performance caused by EEG temporal correlation and guide the development of BCI in a positive direction.

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

# A Appendix A

## A.1 Photography of the watermelon subject

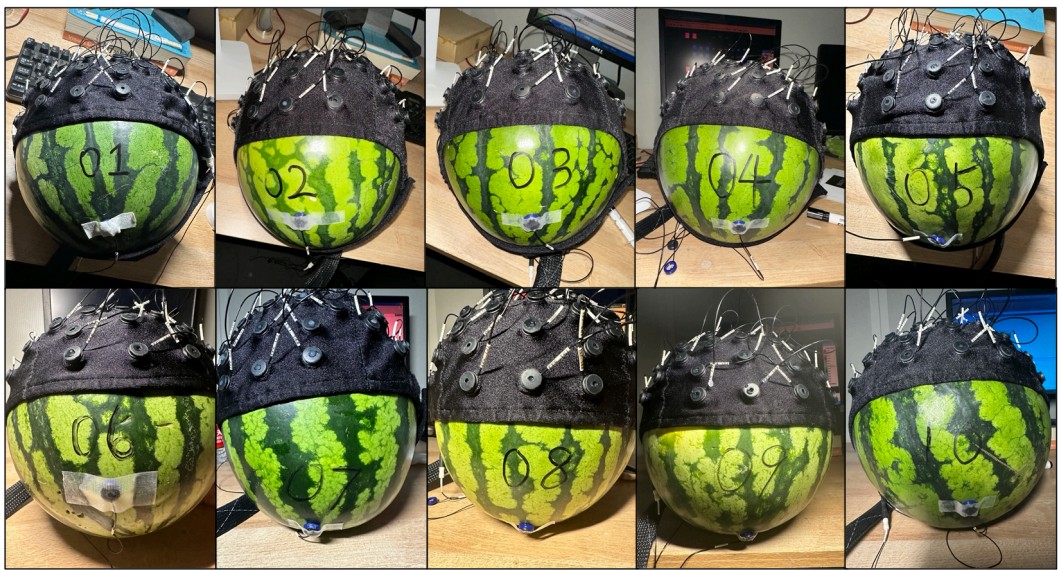

Figure 4: Photos of watermelons used in the experiment. Each watermelon's ID is marked on the watermelon, with IDs ranging from 1 to 10.

## A.2 Reorganization for KUL dataset and DEAP dataset

For the emotion recognition task, we referred to the experimental design of DEAP dataset [37]. In this dataset, the EEG data were recorded while subjects are presented with 40 audio-visual clips of 60 seconds in length, with each corresponding to one of four emotion classes. We only used the first 32 channels of the EEG to match the EEG channel numbers in the DEAP dataset. The watermelon EEG data and SparrKULee EEG data were down-sampled to 128 Hz and then were segmented into 40 60-second segments. The interval between adjacent segments is set to 40 seconds to match the rest time of the subjects during the EEG recording in the KUL dataset. Each segment was assigned a unique domain label and a class label in accordance with the DEAP dataset, and each segment was further segmented into 2-second samples [25]. The reorganized datasets for the Watermelon EEG Dataset and SparrKULee Dataset are called WM-DEAP and SK-DEAP, respectively.

For the ASAD task, we referred to the experimental design of the KUL dataset [38]. In this dataset, 8 clips of two-talker mixed speech are presented to subjects, with each lasting for 6 minutes. Each speech clip contains a left talker and a right talker. Subjects are instructed to attend left or right talker during the entire duration of one clip presentation. The watermelon EEG data and SparrKULee EEG data were down-sampled to 128 Hz and then were epoch into 8 6-minute segments. The interval between adjacent segments is set to 1-2 minutes to match the rest time of the subjects during the EEG recording in the KUL dataset. Each segment was assigned a unique domain label and a class label in accordance with the KUL dataset and was further segmented into 1-second samples [22, 21, 23]. The reorganized datasets for Watermelon Dataset and SparrKULee Dataset are called WM-KUL and SK-KUL, respectively.

## A.3 Detailed implementation of joint training

The joint training was performed on the WM-CVPR and SK-CVPR datasets. All EEG samples were randomly divided into the training set, validation set, and test set in a ratio of 8:1:1. The image encoder of the CLIP (CLIP VIT-L/14) model [1] is chosen to extract image representation, yielding

---

[1] https://huggingface.co/openai/clip-vit-large-patch14

768-dimensional vectors from the image inputs. The structure of the EEG encoder is similar to the model introduced in Subsection 2.3, with an augmentation from 40 to 768 output nodes to match the dimension of the image representation. The network is trained using either a cosine similarity (CS) loss or an InfoNCE contrastive loss (with a temperature parameter set to 0.07). The evaluation metrics selected are Top-1 accuracy, Top-5 accuracy, and Rank accuracy, where the Top-1 accuracy metric is equivalent to the classification accuracy in the classification task.

## A.4  Detailed implementation of image generation

We take an approach similar to previous works [44] [2]. We used a CLIP image encoder to extract image representation and trained an EEG encoder with cosine similarity loss to reconstruct image representation from EEG. This process is the same as described in Joint training with image features. The reconstructed features are then serviced as a conditional input of an image generator. To match the reconstructed features, we employ the pre-trained StableDiffusion model [3] as our generator. This model uses a fixed pre-trained image encoder (CLIP VIT-L/14) to extract image features, which then guide the Latent Diffusion models generation process in the latent space. The diffusion model gradually generates images from a random noise distribution that corresponds to the conditional features during its iterative process. To improve the generation performance, we fine-tuned the generator with the reconstructed image features and the corresponding images. Experiments were done on the WM-CVPR and SK-CVPR datasets. All EEG samples were randomly divided into training set, validation set, and test set in a ratio of 8:1:1.

Consistent with previous work [1], we evaluate the semantic correctness of the generated images using N-way Top-1 and Top-5 accuracy classification tasks. Specifically, given a generated image input, a pre-trained ImageNet1K classifier is used to output a classification logit probability among 1000 classes. Among the 1000 classes, N-1 random classes and the correct class are selected, and the Top-1 and Top-5 classification accuracy are calculated. To avoid randomness, this operation is repeated 50 times for each generated image, with the average value taken as the accuracy.

## A.5  leave-subjects-out data splitting strategy

In this subsection, we employed the leave-subjects-out data splitting strategy. This refers to using data from a subset of subjects for training, while data from the remaining subjects are used for testing. Within the training data, there are two further data partitioning methods: leave-samples-out and leave-subjects-out. The former involves randomly dividing all samples of the training data into training and validation sets, whereas the latter uses data from a subset of subjects for the training set, with the remaining subjects data allocated for the test set. Table 5 presents the decoding accuracy for six datasets (i.e., WM-CVPR, WM-DEAP, WM-KUL, SK-CVPR, SK-DEAP, and SK-KUL).

It can be observed that when the leave-samples-out splitting strategy was used within the training data, both the training and validation sets achieved very high decoding accuracy, but the accuracy only reached the chance level on the test set. Such results are similar to those reported by [46, 47, 8], which corroborates the argument that while the leave-subjects-out approach may avert the domain features leakage, it cannot prevent overfitting of the domain features during the training stage, as discussed in Subsection 4.1. Moreover, when the leave-subjects-out data splitting strategy was used within the training dataset, the validation set performance was only at chance level despite high accuracy on the training set. This further demonstrates that decoding that relies on domain features cannot be generalized to practical application scenarios.

---

[2]https://github.com/bbaaii/DreamDiffusion
[3]https://huggingface.co/runwayml/stable-diffusion-v1-5

Table 5: Decoding accuracy (%) for the six datasets on training, validation and test set. Leave-subjects-out data splitting strategy is used for training and test data. Leave-samples-out and leave-subjects-out data splitting strategy is used for training and validation set. The mean accuracy and standard deviation are calculated over subjects level with a five-fold cross-validation.

| Data splitting strategy for validation set | | WM-CVPR | WM-DEAP | WM-KUL | SK-CVPR | SK-DEAP | SK-KUL |
|---|---|---|---|---|---|---|---|
| leave-samples-out | Training | $80.93 \pm 1.68$ | $87.86 \pm 1.48$ | $99.54 \pm 0.16$ | $69.17 \pm 1.03$ | $76.22 \pm 0.71$ | $100.00 \pm 0.00$ |
| | validation | $80.55 \pm 1.59$ | $86.10 \pm 1.63$ | $99.43 \pm 0.24$ | $68.86 \pm 1.20$ | $74.55 \pm 0.60$ | $100.00 \pm 0.00$ |
| | Test | $2.46 \pm 0.16$ | $24.22 \pm 0.48$ | $48.37 \pm 2.15$ | $2.70 \pm 0.63$ | $26.71 \pm 0.87$ | $50.22 \pm 1.14$ |
| leave-subjects-out | Training | $78.93 \pm 1.09$ | $86.40 \pm 0.75$ | $99.59 \pm 0.16$ | $72.31 \pm 0.59$ | $77.43 \pm 0.52$ | $100.00 \pm 0.00$ |
| | validation | $3.70 \pm 0.34$ | $22.23 \pm 1.29$ | $56.13 \pm 3.06$ | $4.15 \pm 0.60$ | $24.57 \pm 0.33$ | $53.24 \pm 2.85$ |
| | Test | $2.26 \pm 0.16$ | $24.90 \pm 0.43$ | $52.06 \pm 1.26$ | $2.13 \pm 0.29$ | $25.61 \pm 0.43$ | $45.22 \pm 2.83$ |
| | Chance level | 2.50 | 25.00 | 50.00 | 2.50 | 25.00 | 50.00 |

## A.6 Results on different frequency band

To demonstrate that domain features are not solely due to baseline drift, we conducted an analysis on seven frequency bands across six datasets. These seven frequency bands are delta (0-4 Hz), theta (4-8 Hz), alpha (8-12 Hz), beta (12-32 Hz), low gamma (32-45 Hz), and high gamma (55-95 Hz). High gamma frequency band results for DEAP and KUL datasets are not presented due to the sampling rate of 128 Hz (i.e., only frequency under 64 Hz is available according to the Nyquist sampling theorem). Tables 6, 7, 8, and 9 show the decoding accuracy for domain label classification (DLC-EEG), class label classification from domain features (TLC-DF), class label classification directly from EEG (TLC-EEG), and class label classification directly from EEG when samples in the training set and test set are from different domains (TLC-EEG-woDO), respectively. As expected, the highest decoding accuracy is observed for both the low-frequency band (delta band) and the full-frequency EEG data. However, other frequency bands also exhibited decoding accuracy significantly higher than the chance level. This suggests that baseline correction through filtering does not eliminate domain features. Consequently, any experimental designs and data partitioning strategies that could lead to the leakage of domain information should be meticulously avoided.

Table 6: Decoding accuracy (%) using different EEG bands for domain label classification (DLC-EEG)

| | WM-CVPR | WM-DEAP | WM-KUL | SK-CVPR | SK-DEAP | SK-KUL |
|---|---|---|---|---|---|---|
| Full | $88.78 \pm 4.95$ | $96.98 \pm 0.76$ | $99.99 \pm 0.01$ | $69.83 \pm 2.98$ | $72.70 \pm 1.36$ | $100.00 \pm 0.00$ |
| Delta | $88.58 \pm 5.11$ | $96.31 \pm 0.89$ | $99.99 \pm 0.01$ | $69.65 \pm 2.88$ | $72.76 \pm 1.24$ | $100.00 \pm 0.00$ |
| Theta | $8.90 \pm 1.95$ | $10.54 \pm 2.17$ | $41.97 \pm 5.50$ | $11.24 \pm 1.60$ | $10.19 \pm 1.15$ | $43.11 \pm 5.13$ |
| Alpha | $8.62 \pm 1.77$ | $12.88 \pm 2.80$ | $43.42 \pm 5.96$ | $15.16 \pm 1.76$ | $12.87 \pm 1.00$ | $47.67 \pm 4.70$ |
| Beta | $18.53 \pm 3.18$ | $18.18 \pm 2.72$ | $57.85 \pm 4.86$ | $43.95 \pm 2.27$ | $43.68 \pm 1.97$ | $97.17 \pm 0.71$ |
| Low gamma | $39.74 \pm 7.35$ | $62.59 \pm 5.95$ | $85.97 \pm 2.82$ | $53.72 \pm 2.25$ | $52.82 \pm 1.40$ | $96.57 \pm 0.96$ |
| High gamma | $42.15 \pm 7.39$ | - | - | $61.55 \pm 1.94$ | - | - |
| Chance level | 2.50 | 2.50 | 50.00 | 2.50 | 2.50 | 50.00 |

Table 7: Decoding accuracy (%) using different EEG bands for class label classification from domain features (TLC-DF)

| | WM-CVPR | WM-DEAP | WM-KUL | SK-CVPR | SK-DEAP | SK-KUL |
|---|---|---|---|---|---|---|
| Full | - | $92.77 \pm 1.31$ | $100.00 \pm 0.00$ | - | $76.19 \pm 1.80$ | $100.00 \pm 0.00$ |
| Delta | - | $92.12 \pm 1.49$ | $100.00 \pm 0.00$ | - | $76.51 \pm 1.74$ | $100.00 \pm 0.00$ |
| Theta | - | $31.39 \pm 1.80$ | $67.78 \pm 3.56$ | - | $32.17 \pm 1.16$ | $69.41 \pm 3.80$ |
| Alpha | - | $33.10 \pm 2.43$ | $68.78 \pm 4.03$ | - | $33.88 \pm 0.69$ | $71.98 \pm 3.47$ |
| Beta | - | $39.03 \pm 2.09$ | $77.33 \pm 3.71$ | - | $56.91 \pm 2.02$ | $97.83 \pm 0.72$ |
| Low gamma | - | $59.32 \pm 5.22$ | $88.23 \pm 2.56$ | - | $63.80 \pm 1.43$ | $97.44 \pm 0.88$ |
| High gamma | - | - | - | - | - | - |
| Chance level | - | 25.00 | 50.00 | - | 25.00 | 50.00 |

Table 8: Decoding accuracy (%) using different EEG bands for class label classification directly from EEG (TLC-EEG)

|  | WM-CVPR | WM-DEAP | WM-KUL | SK-CVPR | SK-DEAP | SK-KUL |
|---|---|---|---|---|---|---|
| Full | $88.78 \pm 4.95$ | $88.74 \pm 3.26$ | $82.74 \pm 6.44$ | $69.83 \pm 2.98$ | $74.44 \pm 2.76$ | $93.34 \pm 2.01$ |
| Delta | $88.58 \pm 5.11$ | $88.60 \pm 3.36$ | $81.49 \pm 6.44$ | $69.65 \pm 2.88$ | $74.90 \pm 2.55$ | $92.90 \pm 2.15$ |
| Theta | $8.90 \pm 1.95$ | $29.36 \pm 1.27$ | $66.40 \pm 3.47$ | $11.24 \pm 1.60$ | $30.62 \pm 1.30$ | $65.28 \pm 3.83$ |
| Alpha | $8.62 \pm 1.77$ | $31.00 \pm 1.70$ | $68.16 \pm 3.59$ | $15.16 \pm 1.76$ | $32.17 \pm 1.10$ | $67.11 \pm 3.83$ |
| Beta | $18.53 \pm 3.18$ | $35.95 \pm 1.12$ | $71.24 \pm 4.16$ | $43.95 \pm 2.27$ | $43.95 \pm 1.78$ | $93.27 \pm 1.52$ |
| Low gamma | $39.74 \pm 7.35$ | $52.05 \pm 4.72$ | $73.42 \pm 5.37$ | $53.72 \pm 2.25$ | $46.81 \pm 1.03$ | $93.51 \pm 2.02$ |
| High gamma | $42.15 \pm 7.39$ | - | - | $61.55 \pm 1.94$ | - | - |
| Chance level | 2.50 | 25.00 | 50.00 | 2.50 | 25.00 | 50.00 |

Table 9: Decoding accuracy (%) using different EEG bands for class label classification directly from EEG when samples in the training set and test set are from different domains (TLC-EEG-woDO)

|  | WM-CVPR | WM-DEAP | WM-KUL | SK-CVPR | SK-DEAP | SK-KUL |
|---|---|---|---|---|---|---|
| Full | - | $24.67 \pm 2.31$ | $49.97 \pm 4.67$ | - | $25.34 \pm 1.85$ | $59.32 \pm 4.07$ |
| Delta | - | $25.89 \pm 2.58$ | $49.72 \pm 4.85$ | - | $24.71 \pm 1.74$ | $58.25 \pm 3.76$ |
| Theta | - | $23.91 \pm 0.63$ | $49.10 \pm 3.13$ | - | $23.28 \pm 2.18$ | $51.89 \pm 4.32$ |
| Alpha | - | $23.50 \pm 0.82$ | $49.70 \pm 2.91$ | - | $23.26 \pm 1.68$ | $52.77 \pm 4.04$ |
| Beta | - | $22.96 \pm 1.25$ | $50.30 \pm 4.35$ | - | $24.21 \pm 1.39$ | $57.32 \pm 5.26$ |
| Low gamma | - | $26.75 \pm 2.17$ | $49.46 \pm 3.63$ | - | $25.72 \pm 1.61$ | $54.88 \pm 4.92$ |
| High gamma | - | - | - | - | - | - |
| Chance level | - | 25.00 | 50.00 | - | 25.00 | 50.00 |

## A.7 LRTC

The autocorrelation analysis was used to evaluate long range temporal correlation in EEG data from the Watermelon and SparrKULee datasets, similar to the approach taken by previous study. For a lengthy segment of single-channel EEG, the Morlet wavelet transform was employed to extract the time-varying amplitude envelope $W_f(t)$ at a given frequency $f$. The autocorrelation function $ACF_f$ for $W_f(t)$ is defined as:

$$ACF_f(\tau) = corr(W_f(t), W_f(t + \tau)) \tag{4}$$

In the above equation, $corr(,)$ denotes the Pearson correlation coefficient between two time series, and $\tau$ represents the time lag.

In our analysis, the original EEG data were down-sampled to 200 Hz. Ninety-five analysis frequencies were distributed linearly and evenly between 1-95 Hz. Two hundred autocorrelation time lags were logarithmically spaced between 0.5 s and 500 s. For each subject in the Watermelon dataset, continuous EEG recordings were divided into five segments of equal length (with each segment ranging from 15 to 20 minutes), and autocorrelation analysis was completed on each segment. For each subject in the SparrKULee dataset, the autocorrelation analysis was carried out separately on each of their ten trials. Figure 5 shows the results of the autocorrelation analysis for the Watermelon and SparrKULee datasets. The figure illustrates the magnitude of correlation at different frequencies and time lags (represented by color). The correlation values were obtained by averaging the results across all subjects, segments (trials), and electrodes. Black lines represent the contour lines where $p = 0.01$, as determined by statistical analysis. Statistical significance was assessed using single-sample t-test at the subject-electrode level. Specifically, for each electrode of each subject, the averaged Pearson correlation coefficient across all segments (trials) was used as the value for the t-test. Additionally, $p$-values were corrected for multiple comparisons using the Benjamini-Hochberg False Discovery Rate (BH-FDR) to type I error.

As demonstrated in Figure 5, EEG data from both Watermelon and SparrKULee datasets show significant LRTC across multiple frequency bands. For the EEG data from the Watermelon dataset, significant bands of LRTC are primarily distributed in the low-frequency range (<8 Hz) and around 50 Hz, with these correlations spanning over 500 seconds. This indicates that baseline drifts and line

noise contribute to the temporal correlation observed in the Watermelon dataset. For the EEG data from the SparrKULee dataset, LRTCs are significant across the entire frequency range. Similarly, LTRCs are most prominent at low frequencies (<5 Hz) and around 50 Hz, consistent with the findings from the Watermelon dataset. Notably, for SparrKULee dataset, there is also a significant presence of LTRC around 10 Hz, which aligns with previous research findings [13], suggesting the temporal correlation of alpha oscillations in human subjects.

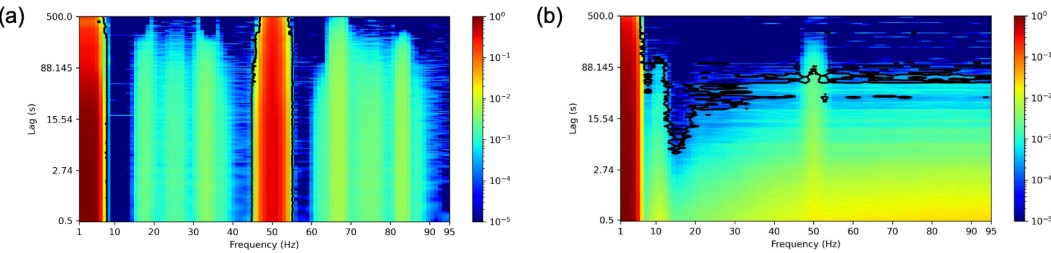

Figure 5: Autocorrelation analysis result on (a) Watermelon and (b) SparrKULee datasets.

