# OpenReview forum: "Beware of Overestimated Decoding Performance Arising from Temporal Autocorrelations in Electroencephalogram Signals"
_NeurIPS.cc/2024/Conference — Submitted to NeurIPS 2024_

### Official Review · Reviewer_S3fr · 2024-07-11

**Soundness:** 4
**Presentation:** 2
**Contribution:** 3
**Rating:** 7
**Confidence:** 5

**Summary:**

The paper highlights how temporal autocorrelations in EEG data can lead to misleadingly high decoding accuracy in brain-computer interface (BCI) tasks. Using a novel approach with a "watermelon EEG dataset," the authors demonstrate that many reported high performances may exploit these autocorrelations rather than genuine neural activity. They propose a unified framework to address this issue across various EEG tasks and recommend improved experimental designs and data splitting strategies to ensure more accurate and reliable results in BCI research.

**Strengths:**

1. Novel Problem Formulation: The paper introduces a novel problem formulation by addressing the potential overestimation of decoding performance in EEG-based brain-computer interfaces (BCIs) due to temporal autocorrelations. This is an innovative perspective that has not been extensively explored in prior research.

2. Creative Use of Non-Human Subjects: The use of watermelons as a model to eliminate stimulus-driven neural responses is highly original. This approach allows for the isolation of temporal autocorrelation effects in EEG signals, providing a unique method to investigate the problem.

3. Impact on BCI Research: The findings have significant implications for BCI research, highlighting a critical issue that could affect the validity of many existing studies. By identifying and addressing this pitfall, the paper provides good insight for more accurate and reliable BCI systems.

**Weaknesses:**

Plz go and check questions.

**Questions:**

1. The paper mentions using a "watermelon EEG dataset" to eliminate stimulus-driven neural responses. What is the scientific rationale and justification for this choice? Why were watermelons chosen over potentially more appropriate models? Have other studies validated the effectiveness of this method?

2. The authors claim that temporal autocorrelations lead to overestimated decoding performance but does not provide a detailed explanation of the specific mechanisms and extent of this impact. How do temporal autocorrelations affect different tasks (e.g., emotion recognition) specifically? Are there quantifiable metrics used to assess this impact?

3. Reproducibility of Experimental Design: The experimental design and data splitting strategies described in the manuscript—are they reproducible across different datasets and experimental conditions? For instance, have similar phenomena been observed with other types of EEG data, such as motor imagery EEG data? Are there specific experimental results supporting this generalizability?

**Limitations:**

While the authors recommend avoiding certain data splitting strategies, the practical implications and feasibility of implementing alternative strategies in real-world BCI applications are not fully explored.

---

> ### Author Rebuttal · Authors · 2024-08-07
>
> We greatly appreciate your careful review and constructive suggestions.
> We are pleased that you mentioned
> "The findings have significant implications for BCI research,
> highlighting a critical issue that could affect the validity of many existing studies",
> as this was the initial objective of our work.
>
>
> **Q1: Questions about phantom EEG**
>
> We have already addressed this issue in the __Author Rebuttal__.
> We hope that it has answered your question.
>
> **Q2: Explanation of the specific mechanisms of temporal autocorrelations. How do temporal autocorrelations affect different tasks (e.g., emotion recognition) specifically? Are there quantifiable metrics used to assess this impact?**
>
> This question encompasses three separate questions, each of which we will reply individually.
>
>
> The specific mechanisms behind temporal-autocorrelations (TA) induced overestimated decoding performance was analyzed and discussed in Appendix A.7, on human and phantom EEG.
> For phantom EEG that records only noise, the significant TA were found at low frequencies and power line frequencies of power spectra.
> For human EEG, the TA were observed across various frequencies.
> Those results indicated the presence of TA in both noise and neural activity.
> Importantly, the TA decay over time, following power-law scaling behavior.
> As a result, when two continuous segments of EEG with different class labels are segmented into multiple samples,
> the similarity within-class samples (which are temporally closer) will be higher than the similarity between-class samples,
> adding each segment samples a unique "domain feature".
> Classifiers would associate the class label with the domain features (as shown in Figure 1 and Figure 2 in the attached PDF), leading to overestimated decoding performance.
>
> As mentioned earlier, TA always exist in EEG, which add unique domain features to continuous EEG segments with the same class label.
> However, due to different experimental designs in various BCI tasks, TA have different effects on decoding.
> In some BCI tasks, such as motor imagery and image decoding, rapid-design paradigms can be used to switch class labels frequently in a short time (e.g., a few seconds or less),
> such that each trial is treated as one sample and temporally adjacent samples mostly have different class labels (unless by chance, adjacent samples have the same label).
> As there is no mapping between domain labels and class labels among those samples,
> the model will not overfit to domain features even during training.
> In contrast, in some BCI tasks, such as emotion recognition and ASAD, requiring subjects to switch class labels every few seconds may not be reasonable.
> In addition, in some BCI tasks, such as sleep stage classification and seizure detection,
> the switch frequency of class label is uncontrollable.
> In both of these latter cases, an experimental condition usually lasts for minutes.
> Continuous EEG segments with the same label need to be split into many samples during training and testing.
> Even when using reasonable data splitting strategies such as leave-subjects-out,
> the model could still utilize domain features to distinguish different classes during the training stage,
> thereby interfering with the learning of class-related features.
>
> Due to the coupling of domain features and class-related features in an actual EEG dataset,
> it is challenging to precisely quantify the impact of TAs on decoding tasks.
> We will acknowledge this limitation in section 4.3.
>
> **Q3: Reproducibility of Experimental Design**
>
> The experimental design and data splitting strategies described in the manuscript are reproducible across different datasets and experimental conditions.
> As described in the __Author Rebuttal__,
> we further added two datasets for two new tasks: the BCIIV2a dataset for motor imagery decoding and the SIENA dataset for epilepsy detection task.
> The high decoding pitfalls were observed on SIENA dataset.
> However, on BCIIV2a dataset, the pitfalls were absent due to the use of rapid-design paradigm during EEG recording, which was in line with the explanation in the __Reply to Q2__.
> This generalizability was supported by the autocorrelation analysis which suggested that the TA features were always existing in EEG data (as shown in the Appendix A.7).
>
> **L1: Aternative strategies in real-world BCI applications (Limitations)**
>
> In the submitted version, we did not clearly describe the implications of our work.
> Our work indicated the necessity of reducing the impact of EEG TA on BCI decoding and give some suggestions on experimental design and model framework construction (detailed in __Author Rebuttal__).
> We will emphasize these practical implications further in the revised Discussion section.
>
> We did not explore implementing alternative strategies to prevent models from overfitting to domain features.
> This is indeed a limitation of this paper, as noted in the limitation section.
> In future work, we will consider using domain generalization to mitigate the impact of EEG TA on decoding.

---

> > ### Comment · Reviewer_S3fr · 2024-08-10
> > **To authors**
> >
> > Thank you for the authors' efforts and responses.
> >
> > I have raised my rating to 7.

---

> > > ### Author Response · Authors · 2024-08-12
> > > **Official Comment by Authors**
> > >
> > > Thank you very much for your constructive review and recent reply. We are confident that your review has enhanced the paper's quality.

---

### Official Review · Reviewer_AKrL · 2024-07-12

**Soundness:** 2
**Presentation:** 2
**Contribution:** 3
**Rating:** 6
**Confidence:** 4

**Summary:**

The paper investigates the potential overestimation of decoding accuracy in brain-computer interface (BCI) tasks that utilize EEG signals. The authors address concerns that high reported decoding accuracies may be attributed to the inherent temporal autocorrelation present in EEG signals rather than the actual decoding of neural responses to stimuli. It contributes to the field of BCI by identifying a potential source of bias in decoding performance, providing a novel dataset to study this issue, and emphasizing the need for careful experimental design to ensure the robustness and reliability of BCI systems.

**Strengths:**

1. This article explores the issue of Overestimated Decoding Performance Arising from Temporal Autocorrelations and verifies it through experiments, with both the expressed viewpoint and experimental process offering high enlightenment value to the BCI community.

2. The self-collected Watermelon EEG is interesting. The use of Watermelon EEG dataset to simulate EEG data without neural activity is a good method to isolate the effects of temporal autocorrelations.

3. The paper provides empirical evidence through experiments that show high decoding accuracies can be achieved even with non-neural datasets, suggesting that reported accuracies in BCI might be influenced by factors other than the models' ability to interpret neural information. The experiment is solid.

**Weaknesses:**

1. This article only covers image decoding, emotion recognition, and ASAD tasks, and to further substantiate the viewpoint presented in this paper, the use of more other tasks or datasets is recommended.

2. The presentation still needs improvement, such as Figures 1 and 2. Some technical terms may be ambiguous, such as “domain”, and should be given more rigorous and clear definitions.

3. The paper only uses a simple CNN (or some parts of this CNN) for EEG classification. A broader range of model testing (e.g. EEGNet and EEG Conformer) would contribute to enhancing the reliability of the research presented in this paper.

**Questions:**

Please see the weakness.

**Limitations:**

The authors have addressed some limitations.But there are still some questions. Please see the weakness.

---

> ### Author Rebuttal · Authors · 2024-08-07
>
> We greatly appreciate your careful review and constructive suggestions.
> We are pleased that you also mentioned "emphasizing the need for careful experimental design to ensure the robustness and reliability of BCI systems",
> which is exactly what we intended to achieve with this work.
>
> **W1: This article only covers image decoding, emotion recognition, and ASAD tasks, and to further substantiate the viewpoint presented in this paper, the use of more other tasks or datasets is recommended**
>
> We have added two EEG datasets: the BCIIV2a dataset for motor imagery (MI) decoding and the SIENA dataset for epilepsy detection. We should note that in BCIIV2a dataset, researchers usually treat each domain as a sample, making the model not rely TA for decoding. We reorganized Watermelon Dataset and sparKULee Dataset,
> obtaining WM-BCIIV2a, SK-BCIIV2a, WM-SIENA, and SK-SIENA.
> Additionally, we also performed decoding on the five actual datasets: CVPR, DEAP, KUL, BCIIV2a, and SIENA.
> All those results are presented in Table R1 in the attached PDF in the __Author Rebuttal__.
>
> **W2: The presentation still needs improvement, such as Figures 1 and 2. Some technical terms may be ambiguous, such as “domain”, and should be given more rigorous and clear definitions.**
>
> We have added subplot Figure 1d, Table R1, and updated the Figure 2 to better formalize the framework, which are presented in the attached PDF in the __Author Rebuttal__.
>
> Table R2 is added to give a clear definition of some used term.
> Table R3 is added to introduce the specific content of the term "domain" and "class" for each dataset.
>
> We hope that these changes will help improve the presentation.
>
>
> **Table R2: Definition of some used term**
>
> | **Term**                 |                                   **Definition**                                   |
> |--------------------------|:----------------------------------------------------------------------------------:|
> | **Class**                |      Distinct category that represents a  specific EEG experimental condition      |
> | **Class-related feature** |                  EEG patterns arising from experimental condition                  |
> | **Domain**               |            A segment of continuous EEG data with  the same class label             |
> | **Domain feature**       |     EEG patterns of samples in a domain, arising from temporal autocorrelation     |
>
> **Table R3: The specific content of the term "domain" and "class" for each dataset.**
>
> | **Dataset** |                      **Domain**                       |                **Class**                 |
> |-------------|:-----------------------------------------------------:|:----------------------------------------:|
> | **CVPR**    |      One of 40 blocks, each lasting 25  seconds       |        One of 40 image categories        |
> | **DEAP**    |      One of 40 trials, each lasting 60  seconds       |       One of 4 emotion categories        |
> | **KUL**     |       One of 8 trials, each lasting 6  minutes        | Attention to the left or right direction |
> | **BCIIV2a** |      One of 576 trials, each lasting 4  seconds       |    One of 4 motor imagery conditions     |
> | **SIENA**   | One of 4-20 EEG segments, each with  varying duration |        Epileptic or non-epileptic        |
>
>
> **W3: The paper only uses a simple CNN (or some parts of this CNN) for EEG classification. A broader range of model testing (e.g. EEGNet and EEG Conformer) would contribute to enhancing the reliability of the research presented in this paper**
>
> We used a simple CNN for EEG classification to demonstrate that the domain features are easy to be learned,
> even with a single-layer CNN.
> However, we agree that employing more models could be helpful to enhance the reliability of our research.
> We added two classical models, EEGNet [1] and EEG Conformer [2] for the EEG classification tasks.
> As shown in Table R1, these models exhibit results similar to those of the simple CNN,
> except EEGNet's extreme results in CVPR dataset.
> In most cases, the CNN outperformed EEGNet and EEG Conformer.
> We must note that EEGNet and EEG Conformer were not designed for learning domain features.
> On the contrary, they have achieved outstanding performance in some datasets where domain features are not shared between training and testing sets.
> For instance, on the BCIIV2a dataset for motor imagery decoding, EEG Conformer has achieved the highest performance although no EEG preprocessing has been performed.
>
> [1] Lawhern et al., J. Neural Eng., 2018, doi: 10.1088/1741-2552/aace8c.
>
> [2] Song et al., IEEE Trans. Neural Syst. Rehabil. Eng., 2023, doi: 10.1109/TNSRE.2022.3230250.

---

> > ### Comment · Reviewer_AKrL · 2024-08-12
> >
> > Thanks for the authors' efforts made for the rebutal. Most of my concerns have been addressed, and I have increased my score accordingly.

---

> > > ### Author Response · Authors · 2024-08-12
> > > **Official Comment by Authors**
> > >
> > > We sincerely appreciate your review and  it really help improve our paper.

---

### Official Review · Reviewer_hA7i · 2024-07-16

**Soundness:** 1
**Presentation:** 1
**Contribution:** 1
**Rating:** 2
**Confidence:** 5

**Summary:**

The authors have correctly identified a significant issue of numerous hyperbolic or irreproducible results in EEG decoding or classification tasks. However, their evaluation approach of recording signals from electrodes placed on a watermelon needs correction. The authors are advised to consult the definition of EEG, as a watermelon is not a brain and does not generate any electrical signals. Therefore, the recorded electrical noises, even when amplified using equipment typically used for EEG, do not constitute EEG data. In summary, while the authors' intentions were good, the numerous errors in their approach make it unacceptable for publication at a top conference such as NeurIPS.

**Strengths:**

An excellent intention to discuss problems with many overblown EEG decoding publications. Yet the conclusions are obvious and many reputable researchers defend their approaches with leave-one-out-subject evaluations to avoid the obvious issues in training and testing data splitting identified by the authors.

**Weaknesses:**

There were unacceptable errors in using EEG terms since instead some environmental or amplifier Brownian noises were recorded after placing electrodes on a watermelon, which probably acted as an electromagnetic antenna capturing all possible low-frequency noises in a room. The CNN application with data splitting issues is too basic for NeurIPS.

**Questions:**

Why was a questionable watermelon selected with the incorrect EEG label? Could simply shuffling the labels of actual EEG experiments, leading to subsequent overfitting, not demonstrate the authors' hypothesis of overfitting pitfalls in ML approaches?

**Limitations:**

Watermelon cannot produce EEG, even if an EEG amplifier records some electrical noise.
The presented study thus hardly relates to EEG decoding problems but seems to report on obvious issues in machine learning due to erroneous data splitting into training and testing sets, thus making it too trivial for NeurIPS.

---

> ### Author Rebuttal · Authors · 2024-08-07
>
> Thanks for your comments.
> It is a pity that presentation of the key points and the specific contribution of this work was not clear enough for you,
> but we hope our responses to your comments and questions can make it clearer.
>
> **S1: Many reputable researchers defend their approaches with leave-one-subject-out evaluations to avoid the obvious issues in training and testing data splitting.**
>
> The strategy of "leave-one-subject-out" is sort of way to avoid the pitfall,
> as mentioned in the work such as ManyDG [1] and VBH-GNN [2].
> However, due to individual differences, training one model for each subject could be more effective. If the dataset size is comparable, it has been known that training models on the subjects' own data could get better decoding performance for those works that don't rely on EEG temporal autocorrelations (TA) for decoding [3].
>
> Meanwhile, despite avoiding the pitfall, simply adopt the "leave-one-subject-out" strategy can not prevent the overfitting on domain features during training stage (shown in Table 5 and Appendix A.5).
> Our work suggests that the key to reduce the impact of EEG TA on BCI decoding is to decouple class-related features from domain features in actual EEG dataset.
>
> **W1: Unacceptable errors in using EEG terms since instead some environmental or amplifier Brownian noises were recorded.**
>
> The term "watermelon EEG" can be changed to "phantom EEG" in the revised version.
> Please notice that we have introduced the concept of "phantom EEG" when we first mentioned "watermelon EEG" in the Introduction (line 79-80).
> The watermelon is widely used as "phantom head" due to its similar conductivity to human tissue,
> similar size and shape to the human head, and ease of acquisition.
> The noises you mentioned can also be recorded when EEG is recorded with human being subjects.
> More details about "The rationality for using watermelon" could be found in __Author rebuttal__.
>
> **W2: The CNN application with data splitting issues is too basic.**
>
> This work is not about "CNN application with data splitting issues".
> Our work concentrates on the impact of EEG temporal autocorrelations (TA) on various BCI decoding tasks.
> The CNN was chosen as the EEG encoder because its structure is simple enough to show that the EEG TA features are easy to be learned.
> More details about the "The contribution of the present study" can be found in the __Author rebuttal__.
>
> **Q1: Why was a questionable watermelon selected with the incorrect EEG label?**
>
> The reason for selecting a watermelon with the manually selected class label is explained in the reply for __W1__ and
> in the section "The rationality for using watermelon" of __Author rebuttal__.
> The advantage of using phantom EEG is to control the interference from stimuli-driven response and subject-related factors,
> and to focus solely on the TA of the noise in EEG. The similar method has been used in previous neuroscience studies.
>
> **Q2: Could simply shuffling the labels of actual EEG experiments, leading to subsequent overfitting, not demonstrate the authors' hypothesis of overfitting pitfalls in ML approaches?**
>
> As shown of Table R1 in the attached pdf, we added experiments as you suggested.
> For the vast majority of BCI tasks, shuffling the labels of actual EEG experiments could demonstrate the hypothesis of overfitting.
> However, when class labels and domain labels correspond one-to-one,
> simply shuffling the labels of a real EEG dataset cannot illustrate the pitfalls.
> For example, in the CVPR dataset, images of 40 classes are presented sequentially in 40 blocks.
> Shuffling the class labels of these 40 blocks is equivalent to swapping the class labels of these 40 blocks in a classification task,
> which does not affect classification accuracy.
>
> Please note that our purpose is not merely for demonstrating the overfitting of TA features.
> We used a unified framework to describe the mechanism of how TA affect EEG decoding across general BCI tasks
> (detailed in "The contribution of the present study" in the __Author rebuttal__).
> Therefore, using a phantom EEG that is independent of the stimulus, subjects, and experimental paradigm, is a more reasonable choice
> (detailed in "The rationality for using watermelon" in the __Author rebuttal__).
>
> **L1: Watermelon cannot produce EEG, even if an EEG amplifier records some electrical noise. The presented study thus hardly relates to EEG decoding problems but seems to report on obvious issues in machine learning due to erroneous data splitting into training and testing sets, thus making it too trivial for NeurIPS.**
>
> The rationale for using watermelon has been presented in the __Author rebuttal__.
>
> Our study is not about erroneous data splitting in machine learning.
> Instead, we focused on the impact of EEG TA on EEG decoding, providing guidance on EEG experimental design and decoding model framework.
> (detailed in "The contribution of the present study" in the __Author rebuttal__).
>
> Meanwhile, the issue of "erroneous data splitting" is not "obvious".
> The issue of TA in EEG decoding has not been widely recognized.
> Some researchers argued that the effect of TA in EEG data on decoding is negligible and many works that utilized TA features to obtain "overestimated decoding performance" were still published.
> As recently as last month, researchers proposed that TA can be controlled in experiments and their effects are marginal [4].
> They even refused to acknowledge the issues with their data splitting.
>
> [1] C. Yang et al. ManyDG: Many-domain Generalization for Healthcare Applications. In ICLR, 2023.
>
> [2] C. Liu et al. VBH-GNN: Variational Bayesian Heterogeneous Graph Neural Networks for Cross-subject Emotion Recognition. In ICLR, 2024.
>
> [3] Y. Song et al. Decoding Natural Images from EEG for Object Recognition. In ICLR, 2024.
>
> [4] S. Palazzo et al. The effects of experiment duration and supertrial analysis on EEG classification methods. In IEEE Trans. Pattern Anal. Mach. Intell., 2024.

---

> > ### Comment · Reviewer_hA7i · 2024-08-13
> > **Thank you for detailed rebuttal**
> >
> > The reviewer appreciates the detailed rebuttal provided by the authors. While the manuscript seems well-written in terms of machine learning, the key issue highlighted is the lack of "EEG" in the "watermelon/phantom EEG." This absence has led the reviewer to give a very negative evaluation. The problem of EEG amplifier noise, whether from the electromagnetic environment or semiconductor noise, is well-known. Despite the authors' references to similar publications, the fact remains that there is no biological tissue in an electronic circuit in the case of the so-called "phantom EEG"; it simply represents electronic circuit noise. Although the electronic circuit is likely the sole source of these technical artifacts, the current manuscript does not address this issue. The reviewer suggests that if the authors were to remove "EEG" from the title and replace "phantom EEG" in the manuscript with "amplifier electronic circuit noise," this change might be acceptable. However, this would render the contribution trivial, given that these issues are widely recognized in the electrophysiological community, just like volume conductance and limitations of EEG amplifier noise.
> >
> > The reviewer's assessment of the manuscript remains unchanged. However, the reviewer acknowledges the potential value of the content to the machine-learning community. If the remaining reviewers can convincingly demonstrate the necessity and educational value of the publication in the machine learning community after adding a precise discussion about EEG amplifier noise as potentially the main source of those TAs, the reviewer would be willing to withdraw from the evaluation process without changing the negative evaluation. This is because such a contribution from an electrophysiological perspective would appear trivial.

---

> > > ### Comment · Area_Chair_BU8p · 2024-08-13
> > > **Agree with Reviewer hA7i**
> > >
> > > I also am missing what this contributes to the BCI field.     The problem of EEG feature drift over time is well known by quality EEG, cognitive neuroscience, and BCI labs and avoided by a variety of methods including interleaving classes, separating training and test data in time, and leave one subject out.  The problem for the CVPR dataset was well demonstrated by the Siskind lab and others.  (Note this is not even just a problem for EEG,  a computer vision algorithm dealing with pictures taken outside in the same location could experience the same problem with a design such as the CVPR dataset, due to changing light angles and spectrum throughout the day.).  Given the drifts, the blocked design is clearly problematic.  Is the point of the watermelon EEG to demonstrate that electrical signals from a head like surface have feature drift that gives rise to this issue?  If the authors addressed where this came from (amplifier noise, low-pass filtering required before A/D sampling,....), it might be more interesting, but given that it is a known issue with EEG (and other signals that drift over time), I don't see the novel contribution in this paper.

---

> > > > ### Author Response · Authors · 2024-08-13
> > > > **Response to Area Chair BU8p (1/2)**
> > > >
> > > > Thank you very much for your comment. We hope our responses can help solve some questions.
> > > >
> > > > **R1: I also am missing what this contributes to the BCI field.**
> > > >
> > > > We have tried to summarize the contribution in the **Author Rebuttal**, and other reviewers' comments
> > > > may be helpful for clarifying this question. Besides, we would like to provide some additional explanations.
> > > >
> > > > Our main contribution is the proposal of a unified framework to describe the mechanism of how
> > > > temporal autocorrelations (TA) affect EEG decoding across general BCI tasks.
> > > > The previous study (**Siskind lab** and others) had the following limitations:
> > > > - They mainly focused on the analysis of a specific task (mainly on image decoding task), making it
> > > >   difficult to extend their conclusion to other tasks.
> > > > - They only proposed that TA could lead to high decoding accuracy pitfalls, but lacked analysis on
> > > >   the mechanism of TA's impact on BCI decoding.
> > > >
> > > > These limitations can be effectively overcome with the proposed framework and the problem formulation.
> > > > BCI researchers can be aware of the pitfalls arising from TA, which exist in various BCI tasks (not just
> > > > in the image decoding task). They were further guided on how to reduce the impact of TA on decoding (more
> > > > details in **Author Rebuttal** and **Q2** in rebuttal to Reviewer S3fr).
> > > >
> > > > **R2: The problem of EEG feature drift over time could be avoided by a variety of methods.**
> > > >
> > > > Indeed, the mentioned methods (i.e., interleaving classes, separating training and test data in time,
> > > > and leave one subject out) could reduce the impact of TA to some extent. Nevertheless, there are some
> > > > limitations.
> > > > - Interleaving classes: this refers to trials that are temporal adjacent have different class labels.
> > > >   The impact of TA could be avoided only when the trial of EEG was not further segmented into several
> > > >   samples; Otherwise, the overfitting or pitfall would occur (more details in **Author Rebuttal** and **Q2** in rebuttal to Reviewer S3fr).
> > > > - Separating training and test data in time: it is efficient when there is a temporal gap between
> > > >   complete training data and complete test data. For example, the training and test data of the BCIIV2a dataset
> > > >   are recorded in different sessions. However, this method merely avoids the pitfall but could not prevent
> > > >   overfitting on domain features during the training stage in some BCI tasks (e.g., the KUL dataset for ASAD),
> > > >   which were not proposed by previous researches.
> > > > - "Leave one subject out": this method can avoid the pitfall, but still has some limitations shown in **S1**
> > > >   in Rebuttal to Reviewer hA7i.
> > > >
> > > > In summary, these methods are appropriate in specific BCI tasks. All these appropriate conditions
> > > > could be summarized in the proposed “leave-domains-out”, which could avoid the pitfall. However, in some BCI tasks,
> > > > the overfitting to the domain features still exists. The key to reducing the impact of EEG TA on BCI decoding
> > > > is to decouple class-related features from domain features in actual EEG dataset. Once when the coupling
> > > > could not be solved with the experiment design and data-splitting strategy (more details in **Q2** in
> > > > rebuttal to Reviewer S3fr), one possible solution is to use domain-invariant representation learning and
> > > > other domain generalization methods.
> > > >
> > > > **R3: The problem for the CVPR dataset was well demonstrated by the Siskind lab and others.
> > > > Given the drifts, the blocked design is clearly problematic.**
> > > >
> > > > We have already referred to these works in our manuscript (line 44-49).
> > > > Here, we would like to emphasize the significant progress of our work compared to theirs.
> > > >
> > > > Firstly, the works of **Siskind lab** focused only on the image decoding tasks.
> > > > They proposed that block-design should not be used to avoid the pitfalls. However,
> > > > with our problem formulation and proposed framework, we found that the pitfall also exists
> > > > in various BCI decoding task even if block-design was not used, such as for the datasets
> > > > we have mentioned (DEAP for emotion recognition, KUL for ASAD).
> > > >
> > > > Secondly, their viewpoints have not been widely accepted despite the problem was demonstrated
> > > > by the **Siskind lab**. Some researchers believed that TA only plays a marginal role in EEG
> > > > (see the line 49-58 in manuscript), as they argued that the experiment done by the **Siskind lab**
> > > > did not provide subjects with enough rest, leading to stronger TA. As recently as last month,
> > > > they still proposed that TA can be controlled in EEG experiments and they refused to acknowledge
> > > > the issues with the block-design paradigm [1]. This is a key reason for us to use the phantom EEG.
> > > > In this case, all neural signals are removed, but TA still results in overestimated decoding performance.
> > > > This fully illustrates that the conclusion that "TA only plays a marginal role in EEG" is not correct.
> > > >
> > > > [1] S. Palazzo et al. The effects of experiment duration and supertrial analysis on EEG classification methods. In IEEE Trans. Pattern Anal. Mach. Intell., 2024.

---

> > > ### Author Response · Authors · 2024-08-13
> > > **Official Comment by Authors to Reviewer hA7i**
> > >
> > > Thank you for your discussion on using "phantom EEG".
> > >
> > > As you mentioned, the "phantom EEG" records amplifier noise (environment or semiconductor noise). Human EEG, while capturing signals arising from the subject (neural activity, artifacts, etc.), also records these noises. The analysis results on "phantom EEG" indicate that significant temporal autocorrelations (TA) exists even in the noise. We must point out that "phantom EEG" is just one of the datasets we used in this work, and the same analysis was also conducted on datasets of Human EEG (SparrKULee Dataset, see line 142-147 in the manuscript). As an extreme example, the "phantom EEG" was used to decouple neural activities from the recorded Human EEG, rather than to demonstrate the main source of TA. Hence, removing "EEG" from the title is not reasonable. We understand your point that the "phantom EEG" indeed does not contain any "EEG" but records only noise, and we will add the discussion about what is really recorded in "phantom EEG" to avoid confusion in the manuscript.
> > >
> > > BCI is a highly interdisciplinary field in which both electro/magnetic signal analysis and neural decoding methods are crucial. Our current work mainly focuses on neural decoding methods. We present our framework, highlighting the TA-induced pitfalls in current BCI decoding methods, and propose ways to avoid the impact of TA on decoding.

---

> > > > ### Author Response · Authors · 2024-08-13
> > > > **Response to Area Chair BU8p (2/2)**
> > > >
> > > > **R4: Is the point of the watermelon EEG to demonstrate that electrical signals from a head like
> > > > surface have feature drift that gives rise to this issue?**
> > > >
> > > > No. The present work aims to demonstrate that TA significantly impacts various BCI tasks even in
> > > > conditions without neural activity (more details in **Author Rebuttal** and **Official Comment by Authors to Reviewer hA7i**).
> > > > However, it cannot be concluded that all TA arise from the noise.
> > > > Contrary to this notion, the SK dataset, in which the EEG was recorded on humans, exhibits TA in
> > > > broader frequencies (Figure 5 and Appendix A.7).
> > > >
> > > > **R5: Given that it is a known issue with EEG (and other signals that drift over time),
> > > > I don't see the novel contribution in this paper.**
> > > >
> > > > The contribution of the present work has also been demonstrated in the **Author Rebuttal** and **R1** detailly.
> > > > In practical, the proposed unified framework serves as a reminder to BCI researchers of the impact of TA
> > > > on their specific BCI tasks, and is intended to guide them on selecting the appropriate experimental design,
> > > > splitting strategy and model construction. Recently, we have observed that some studies continue to rely on
> > > > the TA to achieve the “overestimated decoding performance” in various BCI tasks, such as image decoding [2-4]
> > > > , emotion recognition [5-7], and ASAD [8-10]. We believe that the present work can alarm researchers
> > > > about the impact of TA, and promote the proper application of machine learning in BCI field.
> > > >
> > > > [2] Z. Imani and M. Ezoji. Multi-level brain-guided fusion to reinforce spiking neural network in image classification. In Multimed Tools Appl, 2024.
> > > >
> > > > [3] T. Mwata-Velu et al. Multiclass Classification of Visual Electroencephalogram Based on Channel Selection, Minimum Norm Estimation Algorithm, and Deep Network Architectures. In Sensors, 2024.
> > > >
> > > > [4] L. Zheng et al. EidetiCom: A Cross-modal Brain-Computer Semantic Communication Paradigm for Decoding Visual Perception. In arXiv preprint arXiv:2407.14936, 2024.
> > > >
> > > > [5] M. Alghanim et al. A Hybrid Deep Neural Network Approach to Recognize Driving Fatigue Based on EEG Signals. In Int. J. Intell. Syst., 2024.
> > > >
> > > > [6] Y. Gao et al. EEG emotion recognition based on data-driven signal auto-segmentation and feature fusion. In J. Affect. Disord., 2024.
> > > >
> > > > [7] Z. Zhang et al. Beyond Mimicking Under-Represented Emotions: Deep Data Augmentation with Emotional Subspace Constraints for EEG-Based Emotion Recognition. In AAAI, 2024.
> > > >
> > > > [8] X. Zeng et al. Attention-guided graph structure learning network for EEG-enabled auditory attention detection. In J. Neural Eng., 2024.
> > > >
> > > > [9] Q. Ni et al. DBPNet: Dual-Branch Parallel Network with Temporal-Frequency Fusion for Auditory Attention Detection. In IJCAI, 2024.
> > > >
> > > > [10] M. A. Tanveer et al. Deep learning-based auditory attention decoding in listeners with hearing impairment. In J. Neural Eng., 2024.

---

### Official Review · Reviewer_7nx7 · 2024-07-28

**Soundness:** 4
**Presentation:** 3
**Contribution:** 3
**Rating:** 8
**Confidence:** 4

**Summary:**

Authors hypothesise that the high temporal correlation of EEG data contributes to the high BCI decoding accuracies reported in some prior BCI studies. Specifically, the highly questionable data partitioning practice of splitting continuous EEG data with the same label (or subject) across train/test sets. They present a framework to assess the impact of temporal correlation of EEG features on three different BCI decoding tasks applied to independent datasets, human and watermelon (phantom) EEG data. The inclusion of watermelon dataset is to separate the influence of stimulus-driven responses from highly correlated temporal EEG features that is not fully eliminated when using human EEG data. Results based on the standard data partitioning show high BCI decoding performance for the various tasks even when using watermelon EEG data, and performance is significantly reduced to around chance level when the impact of temporal autocorrelation is mitigated with alternative data partitioning schemes.

**Strengths:**

**Originality**

-	The inclusion of “phantom EEG” recorded from watermelon to disambiguate stimulus-driven neural responses and temporal autocorrelation during the data analysis, which is not fully eliminated with human EEG recordings.

**Quality**

-	Authors provide a theoretical basis to justify their hypotheses and experiment design plan.
-	Analysis plan includes data partitioning used in different BCI decoding tasks (image classification, emotion recognition, auditory spatial attention) applied to a different BCI task (speech evoked response).

**Clarity**

-	The paper is generally well-written. Problem well illustrated in Figure 1.
-	Some areas require clarity (maybe figures?) to better illustrate the different analyses in the framework (for other applications) and results.

**Significance**

-	Highlights the need for more robust experimental design and data partitioning practices in BCI decoding tasks to minimise the impact of inherent temporal correlations of EEG data on performance.
-	The paper demonstrates a limitation of deep learning models (“black box”) in relation to correlation vs. causation.

**Weaknesses:**

•	Adding the performance of the current framework on the actual datasets (CVPR, DEAP, KUL, if publicly available), as well as additional independent BCI datasets would provide other benchmarks for comparison.

•	Not sure why there is a need to match number of the subjects in the SparrKULee dataset to that of the WM “subjects”. The objective of the study is to provide a framework for exploring the impact of temporal correlation of EEG features on BCI decoding performance, not directly comparing both datasets. So, it is fine to include data from all subjects in SparrKULee database.
> To match the number of subjects in the Watermelon EEG Dataset, EEG data from 10 subjects… from the SparrKULee Dataset were used.

**Questions:**

**Questions**

- Does “chance level” consider the distribution of samples per class?

- Can the authors elaborate on the specifics/differences in the set up for “works that do not rely on EEG temporal autocorrelation features for” BCI task decoding (lines 312-315] vs. those presented in the paper?

- Figure 2: Add more details in caption to understand the content in (task, domain, label, etc.). Same for lines 258-262 in results. Need to include visualisation with/without leave-domains-out during training to support results in Table 1. Same for Table 2/Fig 2C.

**Suggestions**

- Recommend that authors better formalise the framework. Authors can introduce each section in a generalised setting, with the conditions tested used as specific example. This would allow for others to apply the generalised framework to their respective application. - Illustration of the three BCI decoding tasks and data reorganisation would be useful for a non-BCI audience.

- The discussion section introduces new analyses rather discussing the specifics/interpretations of the results presented earlier. While the additional analyses introduced in discussion is highly relevant, the extended presentation is an indication of a different placement. This is so that the discussion focuses more on interpretation/implications of the findings (referencing the relevant results) towards a conclusion of the paper.

- Table 5 (training, validation and test set splits) needs to be moved to the main text as it highlights an important aspect of the impact of data partitioning on performance.

- When introducing the watermelon dataset, authors should clarify that it was collected internally for disclosure. This is mentioned towards the end of the paper and may be missed. Authors should also include documentation in the data repository (zenodo) to describe their dataset (including motivation).

- Describe the dataset reorganisation and decoding task for DEAP and KUL as was done for CVPR (lines 154-164). (Found in Appendix, need to move to main text to enhance understanding.)

- Proofread for typos and grammar (not an exhaustive list)

         o    “researches segment the EEG data”
         o	Table 1: “TCL (chance level)”
         o	“retravel task”
         o	Differentials in equations (2) and (3)
         o	“the class-related feature has none possibility”, “there is none class-related feature”

- Define InfoNCE (Table 3)

- Figure 3: While mentioned in caption, better if images are annotated as target and predicted images (can be at top)

**Limitations:**

Authors acknowledge limitations of their work.

---

> ### Author Rebuttal · Authors · 2024-08-07
>
> # Reviewer1 7nx7
>
> We sincerely appreciate you for the thorough review and constructive comments.
> We would like to express our gratitude to you for the high praise on the originality, quality, clarity, and significance of our work.
>
>
>
> **W1: Adding the result of other BCI datasets and actual datasets**
>
> As shown in Table R1 in the attached PDF,
> we have added the performance of the current framework on the actual datasets
> and two other datasets for motor imagery task and epilepsy detection task.
>
>
>
> **W2: Number of the subjects in the SparrKULee dataset**
>
> We acknowledge that we did not provide a suitable explanation.
> Reorganizing the EEG data from SparrKULee subjects
> into other datasets requires each subject to have sufficient long EEG recordings.
> For example, for the KUL dataset, each subject completed 8 trails, with each trial lasting at least 6 minutes.
> This requires that subjects in the SparrKULee dataset have at least 8 runs of recordings,
> with each run lasting more than 6 minutes.
> When reorganizing to the DEAP and CVPR datasets,
> similar constraints also need to be considered.
> Under these constraints, 29 out of the 85 subjects in the SparrKULee dataset meet the requirements.
> As shown in Table R2, we have conducted the same experiment for all the other 19 subjects (SK19) and obtained similar (maybe higher) results.
> In the updated version, we will use the results of all the 29 participants.
>
> __Table R2: Experiment for SK19 and SK__
>
> | |SK19-CVPR|SK19-DEAP|SK19-KUL|SK-CVPR|SK-DEAP|SK-KUL|
> |:-:|:-:|:-:|:-:|:-:|:-:|:-:|
> |DLC|93.68±1.51|89.69±1.48|100±0.00|69.83±2.98|72.70±1.36|100.00±0.00|
> |DLC (chance level)|2.50|2.50|12.50|2.50|2.50|12.50|
> |TLC-DF|-|85.13±1.91|100.00±0.00|-|76.19±1.80| 100.00±0.00|
> |TLC-EEG|93.68±1.51|84.65±2.67|94.92±1.91|69.83±2.98|74.44±2.76|93.34±2.01|
> |TLC-EEG-woDO|-|25.13±4.39|53.66±7.60|-|25.34±1.85|59.32±4.07|
> |TCL (chance level)|2.50|25.00|50.00|2.50|25.00|50.00|
>
>
>
> **Q1: “chance level”**
>
> The sample distribution per class is balanced.
> The chance level is determined by 1/class_num.
>
>
>
> **Q2: Elaborate on the specifics/differences in the set up for BCI task decoding whether relying on EEG temporal autocorrelation**
>
> As we concluded in the __Author rebuttal__, when a segment of continuous EEG data with the same class label are
> divided into the training set and test set, the model might rely on EEG temporal autocorrelation (TA) features for decoding.
>
> For example, in the CVPR2017 dataset, 50 trials with the same class label are presented consecutively,
> and these trials are randomly divided into the training set and test set.
> Decoding tasks on this dataset rely on EEG TA features.
>
> Conversely, for work NICE-EEG [1], the Things-EEG [2] dataset was used.
> The EEG data used for testing was completely separated in time from the EEG data used for training.
> Segments of continuous EEG data with the same label will be only in the training set or the testing set.
>
> For VBH-GNN [3] and ManyDG [4], the "leave-subjects-out" data splitting strategy was used,
> where training and testing data were from different subjects.
> This also ensured that segments of continuous EEG data with the same label will be only in the training set or the testing set.
>
>
>
> **Q3: More details in caption and visualization**
>
> Thank you very much.
> The updated Figure 2 can be found in the PDF file.
> Lines 258-262 in results will be updated in the new version to give a detailed description of the presented figure.
>
>
>
> **S1: Better formalise the framework**
>
> We have added a subplot Figure 1d, Table R1, and updated the Figure 2 in the attached PDF to better formalize the framework.
> Table R3 is added to introduce the specific content of the term "domain" and "class" for each dataset.
> We hope these changes will help non-BCI audiences better understand our generalized framework.
>
> **Table R3: The specific content of the term "domain" and "class" for each dataset.**
>
> |**Dataset**|**Domain**|**Class**|
> |:-:|:-:|:-:|
> |**CVPR**|One of 40 blocks, each lasting 25 seconds|One of 40 image categories|
> |**DEAP**|One of 40 trials, each lasting 60 seconds|One of 4 emotion categories|
> |**KUL**|One of 8 trials, each lasting 6 minutes|Attention to the left or right direction|
> |**BCIIV2a**|One of 576 trials, each lasting 4 seconds|One of 4 motor imagery conditions|
> |**SIENA**|One of 4-20 EEG segments, each with varying duration|Epileptic or non-epileptic|
>
> **S2: Suggestion about discussion**
>
> The discussion is the limited due to page constraints.
> In the updated version, we will discuss "work that relies on and not rely on EEG temporal correlation" (Q2)
> and add further discussion about the effect of EEG temporal autocorrelations (TA) on EEG decoding
> and how to avoid the effect of EEG TA.
>
>
>
> **S3-S6,S8: Suggestion about Table 5 (S3), information about watermelon dataset (S4), reorganization (S5), proofread for typos and grammar (S6), caption for figure 3 (S8)**
>
> We are delighted to accept these suggestions in the updated version.
>
> **S7: Define InfoNCE**
>
> The InfoNCE loss in a batch is defined as follows:
> $
> \begin{aligned} Loss=\frac{1}{N}\sum_{i=1}^N\log\frac{\exp(z_i\cdot v_i/\tau)}{\sum_{j=1}^{N}\exp(z_i\cdot v_j/\tau)} \end{aligned}
> $
>
> where $N$ is the batch size,
> $z_i$ is the latent representation of EEG $i$ extracted by the EEG encoder,
> $v_i$  is the feature of the image corresponding to EEG $i$,
> and $\tau$ is the temperature parameter.
>
> We will add the definition in the updated version.
>
> [1] Y. Song et al. Decoding Natural Images from EEG for Object Recognition. In ICLR, 2024.
>
> [2] A. T. Gifford et al. A large and rich EEG dataset for modeling human visual object recognition. In NeuroImage, 2022.
>
> [3] C. Liu et al. VBH-GNN: Variational Bayesian Heterogeneous Graph Neural Networks for Cross-subject Emotion Recognition. In ICLR, 2024.
>
> [4] C. Yang et al. ManyDG: Many-domain Generalization for Healthcare Applications. In ICLR, 2023.

---

> > ### Comment · Reviewer_7nx7 · 2024-08-12
> >
> > Author rebuttal included a more defined framework for their analysis, additional visuals, clarity on the number of subjects in the SparrKULee dataset, and results from additional EEG datasets. Authors also updated the dataset repository with more information about the dataset within the anonymous constraint of the submission. Authors should refer to FAIR principles for data sharing (https://www.go-fair.org/fair-principles/), and also include information about packages to read the data files (.cnt). Reviewer concerns are mostly addressed. Revising rating.

---

> > > ### Author Response · Authors · 2024-08-13
> > > **Official Comment by Authors**
> > >
> > > Thank you very much for your constructive suggestions and recent strong support. We will organize the datasets to comply with the FAIR principles and provide the packages to read the data files recently. More information about the datasets will be disclosed after the double-blind review period concludes.

---

### Author Rebuttal · Authors · 2024-08-07

Thanks to all the reviewers for their valuable suggestions and for recognizing our intention to reveal the fatal drawback of using DNN on general BCI decoding tasks.
Although three reviewers rated the contribution of this work as "good",
it is a pity that the other one concluded it in an inappropriate way.
Perhaps the usage of term "watermelon EEG" is not precise enough.
However, please note that this term is not the key point of our work,
and phantom EEG collected from watermelons has been widely used in neuroscience studies (see details below).
The critical issues of reviewers' questions are summarized here, and the four reviewers' comments have been replied one by one.
We hope that you will recognize the true contribution of this work,
and we are willing to answer any questions during the discussion period.
We have attached a PDF file containing new tables and figures referenced in our detailed responses below.

**The contribution of the present study**

This work is not about "CNN application with data splitting issues" (Reviewer hA7i).
Instead of focusing on CNN application, our work concentrates on the impact of EEG temporal autocorrelations (TA) on various BCI decoding tasks.
The CNN was chosen as the EEG encoder because its structure is simple enough to show that the EEG TA features are easy to be learned.
Of course, TA features can also be learned by more complex models,
and we have added the experiments suggested by other reviewers (see details below in "additional experiments") for the further confirmation.

In our framework, we proposed the concept of "domain" to represent the EEG patterns resulted from TA and then used phantom EEG to remove stimulus-driven neural responses for the verification.
The results confirmed that the TA, always existing in the EEG data, added unique domain features to a continuous segment of EEG.
The specific finding is that when the segment of EEG data with the same class label are split into multiple samples,
the classifier will associate the sample's class label with the domain features,
interfering with the learning of class-related features.
This leads to an overestimation of decoding performance for test samples from the domains seen during training,
and results in poor accuracy for test samples from unseen domains (as in real-world applications).

Furtherly, our work suggests that the key to reduce the impact of EEG TA on BCI decoding is to decouple class-related features from domain features in actual EEG dataset,
rather than only to simply adopt the strategy of "leave-one-subject-out" (mentioned by Reviewer hA7i), which is quite limited for practical application of BCI.


**The rationality for using watermelon**

The watermelon was served as phantom head that we have stressed in the paper (line 79).
The term "watermelon EEG" can be changed to "phantom EEG" to avoid confusion.
The usage of phantom head allows researchers to evaluate the performance of neural-recording equipment and proposed algorithms
without the effects of neural activity variability, artifacts, and potential ethical issues.
Phantom heads used in previous studies include digital models [1], human skull [2], artificial physical phantoms [3], and watermelons [4]–[7].
Due to their similar conductivity to human tissue, similar size and shape to the human head, and ease of acquisition, watermelons are widely used as "phantom heads".

Previous studies have shown that EEG signals exhibit TA,
which arises from baseline drift and long-range temporal correlations of neural oscillations.
In many BCI datasets, the domain feature caused by TA and the class-related features driven by the stimulus were coupled.
However, it is highly argued that whether the impact of EEG TA is important,
and whether it should be accounted for decoding,
which is a bone of contention for using DNNs in BCI decoding tasks (see citation 8-10 and 15-18 in the manuscript).
The advantage of adopting phantom EEG is to control the interference from stimuli-driven response and subject-related factors.
And it is firstly found in our work that the phantom EEG exhibit the effect of TA on decoding even when only noise was recorded,
indicating the inherent existence of TA in the EEG data.
As commented by the reviewer S3fr, the framework we proposed and verified with phantom EEG data has significant implications for BCI research.

**Additional experiments of other BCI tasks and actual datasets**

We have conducted the experiments suggested by reviewers (7nx7, AKrL and S3fr) with the BCIIV2a dataset [8] for motor imagery task and the SIENA dataset [9] for epilepsy detection task.
All results for the five actual datasets, shuffled datasets (actual EEG with shuffled labels), and their reorganized datasets are presented in Table R1.
Except CNN, two other classical models suggested by reviewer (AKrL) were also used for the verification, EEGNet [10] and EEG Conformer [11].
The conclusion is consistent with that drawn in the manuscript. All those additional experiments results will be added in our updated version.

[1] Wolters et al., NeuroImage, 2006, doi: 10.1016/j.neuroimage.2005.10.014.

[2] Gavit et al., IEEE Trans. Biomed. Eng., 2001, doi: 10.1109/10.951510.

[3] Oliveira et al., J. Neural Eng., 2016, doi: 10.1088/1741-2560/13/3/036014.

[4] Mandelkow et al., NeuroImage, 2007, doi: 10.1016/j.neuroimage.2007.04.034.

[5] Perentos et al., IEEE Trans. Biomed. Eng., 2013, doi: 10.1109/TBME.2013.2241059.

[6] Schaefer et al., NeuroImage, 2011, doi: 10.1016/j.neuroimage.2010.05.084.

[7] Collins et al., IEEE Trans. Med. Imaging, 1998, doi: 10.1109/42.712135.

[8] Tangermann et al., Front. Neurosci., 2012, doi: 10.3389/fnins.2012.00055.

[9] Detti et al., Processes, 2020, doi: 10.3390/pr8070846.

[10] Lawhern et al., J. Neural Eng., 2018, doi: 10.1088/1741-2552/aace8c.

[11] Song et al., IEEE Trans. Neural Syst. Rehabil. Eng., 2023, doi: 10.1109/TNSRE.2022.3230250.

---

### Decision · Program_Chairs · 2024-09-25

**Decision:**

Reject

**Comment:**

This paper points out the clear problem	with block designed EEG experiments,
most notably the CVPR paper, but also evident in other papers where people take
small training and testing segments of a long block of data of one "label". Specifically
this paper also shows the problem, but features it less, for DEAP and KUL.

The idea behind this paper is worthwhile, but is a little too simple
and well known for the low-acceptance rate NeurIPS conference - I have
reviewed at least a few papers this year that mention the problem.
At this point, the issues raised are obvious or self-explanatory to most
experts in the Cognitive Neuroscience and BCI fields.  While the main lab
that published the problematic object recognition paper may still insist
that the problem is not a major factor, others have long realized the issue.

That said, there are unfortunately still some labs that are not aware
of the problems with these datasets and experimental designs.  I
suggest simplifying the message and sending the paper to a journal or
conference that continues to publish papers where this problem is a
factor.	 Indicating a more comprehensive list of other datasets in
use in the field with the same problem would also be a useful contribution for the field.
I think more focus on the dataset problem than the watermelon analysis method you used
would make the paper simpler and more impactful.